# ALS-linked loss of Cyclin-F function affects HSP90

Alexander Siebert[1], Vanessa Gattringer[1], Jochen H Weishaupt[2], Christian Behrends[1]

The founding member of the F-box protein family, Cyclin-F, serves as a substrate adaptor for the E3 ligase Skp1-Cul1-F-box (SCF)$^{Cyclin-F}$ which is responsible for ubiquitination of proteins involved in cell cycle progression, DNA damage and mitotic fidelity. Missense mutations in *CCNF* encoding for Cyclin-F are associated with amyotrophic lateral sclerosis (ALS). However, it remains elusive whether *CCNF* mutations affect the substrate adaptor function of Cyclin-F and whether altered SCF$^{Cyclin-F}$–mediated ubiquitination contributes to pathogenesis in *CCNF* mutation carriers. To address these questions, we set out to identify new SCF$^{Cyclin-F}$ targets in neuronal and ALS patient–derived cells. Mass spectrometry–based ubiquitinome profiling of *CCNF* knockout and mutant cell lines as well as Cyclin-F proximity and interaction proteomics converged on the HSP90 chaperone machinery as new substrate candidate. Biochemical analyses provided evidence for a Cyclin-F–dependent association and ubiquitination of HSP90AB1 and implied a regulatory role that could affect the binding of a number of HSP90 clients and co-factors. Together, our results point to a possible Cyclin-F loss-of-function–mediated chaperone dysregulation that might be relevant for ALS.

## Introduction

Conjugation of ubiquitin (Ub) to proteins (i.e., ubiquitination) controls many cellular processes by directing its targets to proteasomal degradation or altering the functional properties of its targets in a regulatory manner. These different outcomes are the results of an intricate interplay between different types of Ub modifications and their recognition by distinct Ub-binding proteins. The complexity arises from the fact that proteins can be modified either by single Ub molecules on one or multiple lysines and/or by homotypic or branched Ub chains in which Ub moieties are linked via one or several of their seven lysine residues (K6, K11, K27, K29, K33, K48, and K63) and/or the N-terminal methionine (M1) (1). Ubiquitination involves an enzymatic cascade consisting of three orchestrated steps (2). First, an E1-activating enzyme uses ATP to form a Ub thioester on its active cysteine. Subsequently, this Ub is transferred to an E2 conjugating enzyme yielding an E2~Ub thioester (E2~Ub). Last, an E3 ligase recognizes the substrate and brings it into proximity of the E2~Ub. Depending on the class of E3 ligase, the final step involves either the formation of an E3~Ub thioester before the Ub transfer onto substrates or the direct transfer of Ub from the E2 to the substrate (2, 3). The latter mechanism is used by the family of really interesting new gene (RING) E3 ligases of which Cullin-RING ligases (CRLs) represent the largest subgroup. CRLs are modularly built complexes consisting of one of the seven scaffolding Cullins (e.g., CUL1), the RING-finger protein RBX1 which recruits the E2~Ub and a member of one of the several substrate adaptor families such as the F-box proteins (4).

The founding member of this latter family, FBX1 (also known as FBXO1 or Cyclin-F), uses its F-box to bind to CUL1 via the adaptor SKP1, whereas the cyclin domain of Cyclin-F interacts with ubiquitination targets (5, 6). This is different from other cyclins which use their cyclin domain to bind to cyclin-dependent kinases as part of their signaling function during the cell cycle (5, 7, 8, 9, 10). Nevertheless, the Skp1-Cul1-F-box (SCF)$^{Cyclin-F}$ ligase complex controls cell cycle progression by binding to the substrate adaptor fizzy-related protein homolog (FZR1) and by ubiquitinating the transcription factor E2F7 (11, 12). Moreover, Cyclin-F binds the centriole regulator CP110 and the ribonucleotide reductase RRM2 in a cell cycle-dependent manner and mediates their ubiquitination which targets both proteins for proteasomal degradation and is required for maintenance of mitotic fidelity and genome integrity (13, 14). Besides, SCF$^{Cyclin-F}$ contributes to the regulation of other diverse cellular processes such as DNA damage response and mitotic spindle formation by ubiquitinating Exonuclease 1 and Nucleolar and spindle-associated protein 1 (15, 16).

Cyclin-F has been implicated in several diseases. For example, alterations in Cyclin-F protein levels are linked to tumorigenesis and cancer progression (17, 18). Furthermore, mutations in *CCNF*, the Cyclin-F gene, are associated with amyotrophic lateral sclerosis (ALS) (19, 20, 21). ALS is a rapidly progressing neurodegenerative disorder that clinically presents itself through progressive paralysis caused by upper and lower motor neurons loss, which ultimately leads to respiratory failure and thereby death (22). The molecular mechanisms causing ALS are not fully understood; however, several processes contribute to the disease such as increased oxidative stress, dysbalanced cytoskeleton dynamics, disrupted RNA homeostasis and

[1]Munich Cluster for Systems Neurology (SyNergy), Medical Faculty, Ludwig-Maximilians-University München, Munich, Germany   [2]Division of Neurodegenerative Disorders, Department of Neurology, Medical Faculty Mannheim, Mannheim Center for Translational Neurosciences, Heidelberg University, Mannheim, Germany

Correspondence: christian.behrends@mail03.med.uni-muenchen.de

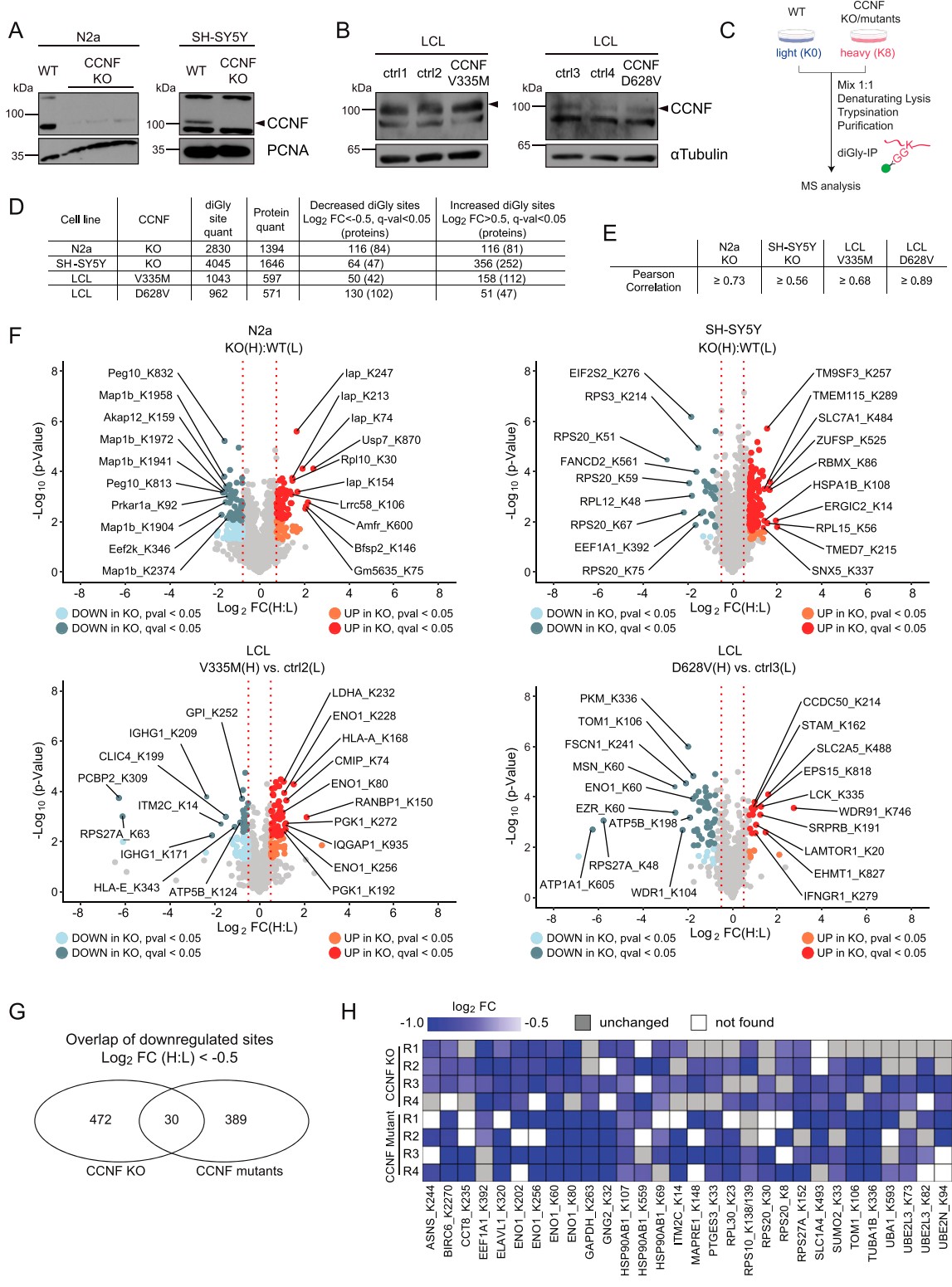

**Figure 1. Ubiquitinome analysis of *CCNF* knockout and mutant cells.**
**(A, B)** Immunoblot analysis of N2a and SH-SY5Y CCNF wild-type (WT) and knockout (KO) cells (A) as well as of lymphoblastoid cells (LCLs) from healthy individuals (ctrl1–4) and amyotrophic lateral sclerosis patients carrying the CCNF mutations V335M and D628V (B). **(C)** Schematic overview of diGly proteomic experiments. Differential SILAC-labeled CCNF WT (N2a, SH-SY5Y) or ctrl (LCLs) and CCNF KO (N2a, SH-SY5Y) or mutant cells (LCLs) were mixed at a 1:1 ratio followed by denaturing lysis, sequential diGly immunoprecipitation, tryptic digestion, desalting, and mass spectrometry analysis. **(D)** Summary of diGly-modified lysines (sites) and proteins from biological replicate experiments (n = 4). Threshold for regulated sites was $\log_2$ fold change (FC) (H:L) > 0.5 or <−0.5 with a q-value < 0.05 (t test). **(E)** Pearson correlation of H:L ratios for

disturbance in protein homeostasis. The latter is manifested by protein aggregation, altered chaperone function and unfolded protein response as well as defects in autophagy and the ubiquitin-proteasome system (UPS) (23). Consistently, overexpression of Cyclin-F carrying the ALS linked S621G mutation was reported to increase ubiquitination in general and in particular ubiquitination of the known SCF$^{Cyclin-F}$ target RRM2 and of the neuropathological ALS marker RNA-binding protein TDP43 in combination with UPS and autophagy impairments (24, 25). In addition, this mutation caused apoptosis activation in different cell models including patient-derived iPSCs and in zebrafish over-expressing Cyclin-F S621G. Intriguingly, these animals showed aberrant neuronal branching and reduced motor function (26, 27). Whereas other ALS mutations in *CCNF* are less well characterized and distributed throughout the *CCNF* gene with no obvious clustering at encoded domains such as the cyclin or PEST domain (20, 21, 24, 28, 29), Cyclin-F S621G served as a paradigm to postulate a gain of toxic function mechanism. Notably, some ALS *CCNF* mutations might perturb cellular proteostasis independent of the SCF substrate adaptor function of Cyclin-F (30). Hence, it remains elusive to what extent altered substrate ubiquitination by SCF$^{Cyclin-F}$ contributes to phenotypic manifestations related to ALS pathogenesis. For the most part, this is due to a lack of knowledge on SCF$^{Cyclin-F}$ targets in neuronal and patient-derived cells.

In this work, we combined quantitative mass spectrometry-based ubiquitin remnant profiling in N2a and SH-SY5Y CCNF knockout cells as well as patient-derived lymphoblastoid cell lines with proximity and interaction proteomics to uncover Cyclin-F ubiquitination targets. Using this approach, we identified the chaperone HSP90AB1 as a new Cyclin-F binding protein which is constitutively ubiquitinated in a Cyclin-F wild-type dependent manner. Importantly, Cyclin-F–mediated ubiquitination of HSP90AB1 regulates the binding of a number of HSP90 clients and co-factors. Overall, our findings indicate that SCF$^{Cyclin-F}$ is required for fine tuning of parts of the cellular chaperone machinery and highlight a role for a loss-of-function mechanism in CCNF ALS.

# Results

### Ubiquitinome analysis of Cyclin-F–deficient cells

To advance our understanding of Cyclin-F malfunctioning in ALS, we set out to identify potential new SCF$^{Cyclin-F}$ substrates in two complementary cellular systems. First, we used CRISPR/Cas9 technology to delete CCNF in two neuron-like cell types, namely murine N2a and human SH-SY5Y (Fig 1A). Second, we used two different ALS patient–derived lymphoblastoid cell lines (LCLs) which expressed Cyclin-F carrying the mutations V335M and D628V.

Whereas V335M is located in the cyclin domain responsible for binding SCF$^{Cyclin-F}$ substrates, D628V is part of the PEST domain which is thought to control the stability of Cyclin-F (31). Notably, both CCNF mutant LCLs did not show overt differences in Cyclin-F protein levels compared with their two respective gender- and age-matched control LCLs (ctrl1-4) carrying wild-type CCNF (Fig 1B). Next, we performed quantitative diGly proteomics to uncover ubiquitination sites which show decreased abundance in this panel of CCNF KO and mutant cell lines and hence represent potential ubiquitination targets of SCF$^{Cyclin-F}$. For this purpose, we combined stable isotope labeling by amino acids in cell culture (SILAC) with immunoaffinity-based enrichment of diGly remnant-containing peptides after tryptic digestion of ubiquitinated proteins. Briefly, CCNF WT and KO N2a and SH-SY5Y cells as well as CCNF WT and mutant LCLs were differentially SILAC labeled, lysed under denaturing conditions and combined in a 1:1 ratio per cell line. After protein extraction and proteolytic digestion, tryptic peptides were subjected to sequential anti-diGly immunoprecipitation (IPs) and analyzed by mass spectrometry (MS) (Fig 1C). In quadruplicate experiments, we quantified a total of 2,830 and 4,045 non-redundant diGly sites in 1,394 and 1,646 proteins in N2a and SH-SY5Y, respectively (Figs 1D and S1A). In LCLs, diGly experiments were performed using four biological replicates for each of the two controls. Therefore, the values reported are the sum of both controls compared with each mutant. The amount of total diGly sites and corresponding proteins was lower in LCLs compared to the KO cell lines with 1,043 and 962 unique diGly sites in 597 and 571 proteins in LCLs expressing CCNF V335M and D628V, respectively (Figs 1D and S2A). Using statistical analysis (*t* test with a q-value of <0.05), we identified between 50 and 130 diGly sites (42–102 proteins) with log$_2$ fold change (FC) < –0.5 in SH-SY5Y, N2a, and LCL cells (Fig 1D and Table S1). Pearson's correlation coefficients between 0.56 and 0.89 indicated high reproducibility between biological replicate samples (Fig 1E). Consistent with Cyclin-F's role in nuclear processes (7, 11, 32, 33), a number of diGly sites that decreased in abundance upon CCNF KO or mutation were found in proteins associated with RNA- or DNA-binding and other nuclear functions (Peg10 in N2a; RPS20, RPL12, and FANCD2 in SH-SY5Y; PCBP2 in CCNF V335M LCLs) (Fig 1F). Other strongly decreased ubiquitination sites were associated with the cytoskeleton which is known to be altered in ALS. For example, five different diGly sites in MAP1B, an α-tubulin binding protein, were found to be down-regulated with a log$_2$ FC < –1.5 in N2a cells, whereas several actin-binding proteins such as MSN, FSCN1, and WDR1 showed decreased ubiquitination in CCNF D628V LCL cells (Fig 1F). Notably, Ub diGly sites representing K63- and K48-linked chains were decreased in both LCLs (RPS27A_K63 in V335M; RPS27A_K48 in D628V) which is different from reports describing elevated Ub K48 levels in cells expressing the ALS-linked Cyclin-F mutation S621G (34). Taking into account that the four cell

---

four respective eight biological replicates in N2a, SH-SY5Y, and LCLs described in Fig 1D. **(F)** Volcano plots depicting relative changes in diGly site abundance for N2a and SH-SY5Y CCNF WT versus KO as well as for representative ctrl versus CCNF mutant LCLs. Significantly decreased or increased diGly sites in CCNF KO and mutant cells were labeled in indicated bright or dark colors representing *P*-value < 0.05 or q-value < 0.05 (*t* test). Top 10 increased and decreased diGly sites with the highest fold change are highlighted. Known CCNF interactors are labeled with black circles. **(G)** Overlap of decreased diGly sites in CCNF KO (N2a and SH-SY5Y) and CCNF mutant conditions (LCLs) with log$_2$ FC < –0.5 in two of four biological replicates. **(H)** Heat map of commonly decreased diGly sites in CCNF KO and mutant cells. Blue scale indicates log$_2$ FC compared with the respective control. Grey boxes mark biological replicates of unchanged diGly sites. White boxes mark biological replicates where the diGly site was not found.

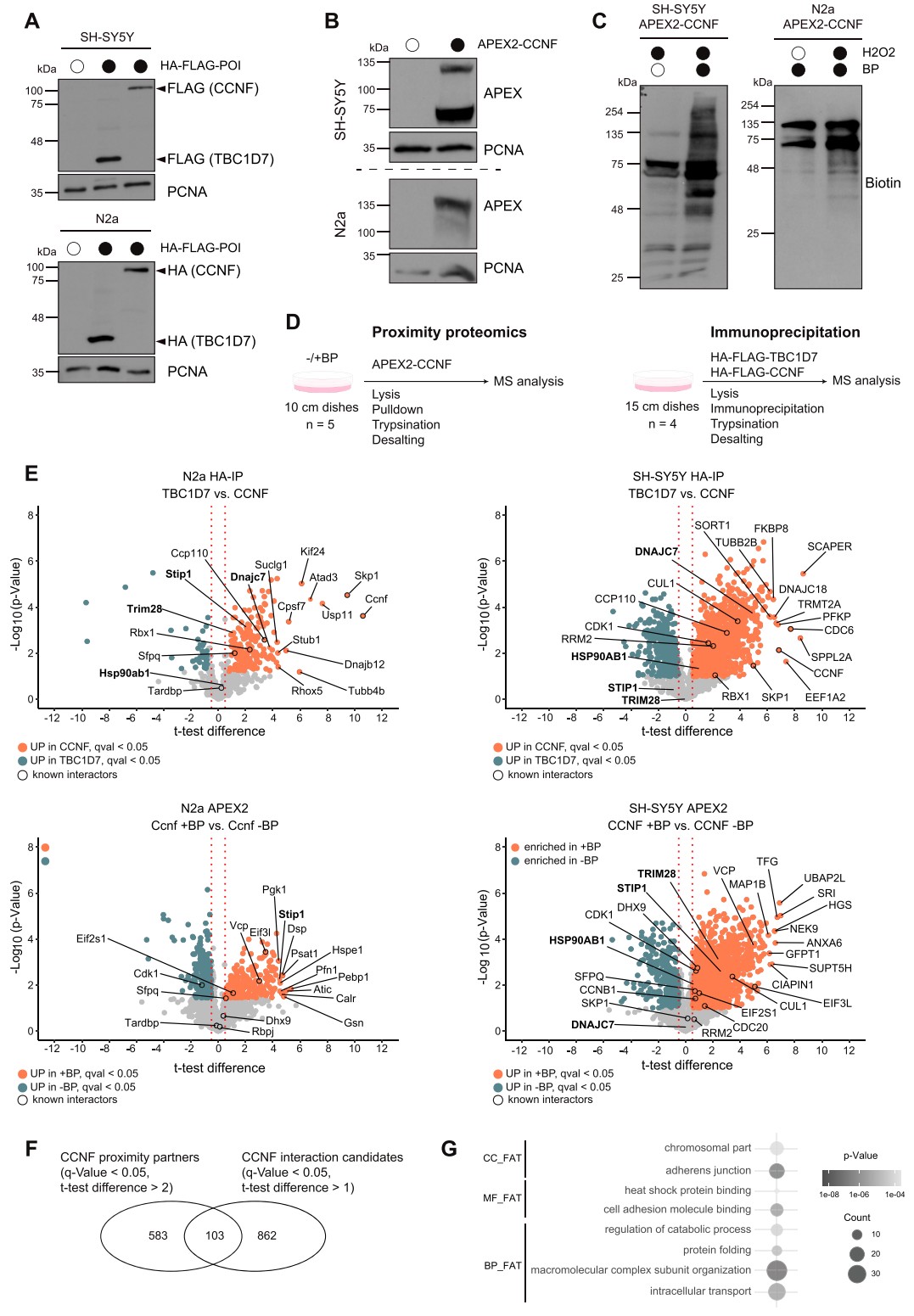

**Figure 2. Cyclin-F proximity and interaction proteomics.**
**(A, B)** Lysates from empty and HA-FLAG-CCNF or -TBC1D7 (A) and APEX2-CCNF (B) overexpressing SH-SY5Y and N2a cells were analyzed by SDS–PAGE and immunoblotting. **(C)** Biotinlyation in SH-SY5Y and N2a cells overexpressing APEX2-CCNF was induced by 30 min biotin-phenol (BP) and 1 min $H_2O_2$ treatment followed by lysis and immunoblot analysis. **(D)** Schematic overview of proximity and interaction proteomics workflow. **(E)** Volcano plots showing changes in abundance of candidate interacting proteins of HA-FLAG-TBC1D7 (control) and HA-FLAG-CCNF in N2a and SH-SY5Y (upper panel) and of proximity partners of APEX2-CCNF in the absence (control) and presence of BP in N2a and SH-SY5Y (lower panel). Proteins enriched in HA-FLAG-CCNF immune complexes or in proximity to APEX2-CCNF are shown in orange

lines are from very different origins and genetic backgrounds, we applied less stringent filtering using $\log_2$ fold change ($\log_2$ FC < −0.5 in at least two biological replicates per each condition) without statistical testing to identify commonly decreased diGly sites in CCNF KO and mutant cells. Using this approach, we found 30 potential ubiquitination sites shared between both CCNF deficiency conditions (Fig 1G). Importantly, protein expression profiling revealed that the vast majority of these potential Cyclin-F ubiquitination targets remained unchanged at the total protein level (Fig S1B and C and Table S2). The corresponding proteins participate in various cellular processes such as RNA (RPL30, RPS10, RPS20, and ELAVL1) and microtubule (MAPRE1) binding as well as ubiquitination (UBA1, UBE2N, and RPS27) (Fig 1H). Of particular interest were diGly sites found in chaperones (HSP90AB1 and CCT8) because disturbance of protein homeostasis is thought to commonly contribute to ALS pathogenesis (23).

### Identification of proximity and interaction partners of Cyclin-F

Complementary to the ubiquitinome profiling approach, we performed proximity biotinylation and IP coupled to MS analysis to identify potential SCF$^{Cyclin-F}$ targets. For this purpose, we generated N2a and SH-SY5Y cell lines, stably expressing Cyclin-F either tagged with a HA-FLAG-tag or fused to a myc tagged version of the engineered ascorbate peroxidase APEX2 (Fig 2A and B). Notably, the functionality of the APEX2 fusion was examined by inducing biotinylation in N2a and SH-SY5Y cells (Figs 2C and S2A). For proximity proteomics cells were grown in the presence of biotin-phenol (BP) for 30 min followed by a 1-min $H_2O_2$-pulse to induce biotinylation. Subsequently, biotinylated proteins were subjected to streptavidin pull-downs. Conversely, interaction proteomics involved enrichment with anti-HA affinity resin and elution with HA peptide. In both cases, samples were digested with trypsin and analyzed by MS (Fig 2D). Expression of HA-FLAG- or APEX2-TBC1D7 or omission of BP was used as negative control conditions. Analysis of Cyclin-F interaction candidates identified in N2a and SH-SY5Y revealed several known interactors such as the SCF$^{Cyclin-F}$ ligase components SKP1, RBX1, and CUL1 as well as a number of their targets including CCP110, RRM2, CDK1, CDC6, and SFPQ (Fig 2E and Table S3). Besides, a large number of proteins were found enriched following Cyclin-F IP compared with the control TBC1D7 IP. Intriguingly, the interaction candidates with the greatest $t$ test difference and a q-value < 0.05 featured a number of functional categories which were also found in the diGly proteomics experiments including chaperones (Dnajc7, Dnajb12, Stub1 in N2a; DNAJC7, DNAJC18, and FKBP8 in SH-SY5Y) and cytoskeleton associated proteins (Kif24, Atad3, Tubb4b, TUBB2B). The proximity partners detected in both cell lines covered a similar molecular landscape with multiple known Cyclin-F–binding partners (CUL1, SKP1, RBX1, CCP110, CDC6, RRM2, VCP, SFPQ, EIF2S1, EIF3L, and DHX9), chaperones (STIP1 HSP90AB1) and cytoskeleton-binding proteins (Myh10, Dsp, Vcl, and MAP1B) (Figs 2E and S2B and Table S4). The fact that we used different cell types and varying experimental conditions (HA-IP versus APEX2) might explain the

observed discrepancies in the detection and scoring of established Cyclin-F–binding partners (Fig S2C). Comparison of both approaches revealed 103 proteins that were commonly found enriched by CCNF proximity and interaction proteomics (Fig 2F). Functional annotation clustering of these shared potential CCNF targets using DAVID yielded gene ontology (GO) terms in accordance to known functions of Cyclin-F in nuclear processes (e.g., chromosomal part) but also functions that were not primarily associated with Cyclin-F such as protein folding and heat shock protein binding (Fig 2G).

### Validation of the HSP90 chaperone machinery as Cyclin-F–binding partner

Taking advantage of our parallel proteomics approaches, we searched the data sets for proteins that were enriched in CCNF's proximitome and interactome (Fig 2F) but carried decreased diGly sites when CCNF was deleted or mutated (Fig 1H). This analysis revealed two new potential Cyclin-F targets, TUBA1A and HSP90AB1 (Fig 3A). Given the role of altered proteostasis in ALS pathogenesis, we focused our subsequent efforts on HSP90AB1. First, we probed for the association of HSP90AB1 with Cyclin-F by IP and immunoblotting. In contrast to the control FBXO28, Cyclin-F showed clear binding to HSP90AB1 (Fig 3B). Intriguingly, two HSP90 co-chaperones DNAJC7 and STIP1 (also known as HOP) present in our proteomics data sets with similar but less prominent features as HSP90AB1 were likewise found to specifically associate with Cyclin-F compared to FBXO28, whereas the CUL1 adaptor SKP1 bound to both F-box proteins (Fig 3B and C). In addition, a number of other interaction candidates such as the Ub-binding protein TOLLIP and the SUMO E3 ligase TRIM28 were also confirmed as Cyclin-F–binding partners (Fig S3A and B). Second, we examined whether the ALS-linked mutations in Cyclin-F affect the binding to HSP90AB1. Thereto, we reconstituted CCNF KO SH-SY5Y cells with wild-type (WT) or mutant (V335M or D628V) HA-FLAG–tagged Cyclin-F and performed HA-IPs. However, Cyclin-F immunoprecipitates showed no overt changes in HSP90AB1 levels across the different Cyclin-F variants (Fig 3D). These findings indicate that HSP90AB1 is a new Cyclin-F interacting protein which binds to Cyclin-F independent of two different ALS-linked CCNF mutations. Notably, the HSP90AB1-Cyclin-F interaction could represent a ligase-substrate or chaperone-client relationship.

### Cyclin-F–dependent ubiquitination of HSP90AB1 regulates its chaperone cycle

To test whether HSP90AB1 is indeed ubiquitinated, as suggested by our diGly proteomics, we performed denaturing IPs with HA-FLAG-HSP90AB1 and HA-FLAG-TBC1D7 as a negative control. Immunoblotting of HA immunoprecipitates with a K48 linkage specific polyUb antibody (Ub-K48) unveiled Ub conjugates on HSP90AB1 which were sensitive to treatment with the deubiquitinase USP2

($t$ test difference > 0.5, q-value < 0.05, FDR-corrected, $t$ test). Top 10 significantly enriched proteins, known CCNF interactors (black circles) and selected candidates are highlighted. **(F)** Overlap between proximity partners (q-value < 0.05, $t$ test difference > 2, $t$ test) and interaction candidates (q-value < 0.05, $t$ test difference > 1, $t$ test). **(E, G)** Gene Ontology (GO) analysis of proteins found in the overlap of (E). Grey gradient represents $P$-values. The number of proteins (count) found associated with a given GO term is indicated by the size of the circles. BP, biological process; CC, cellular component; MF, molecular function.

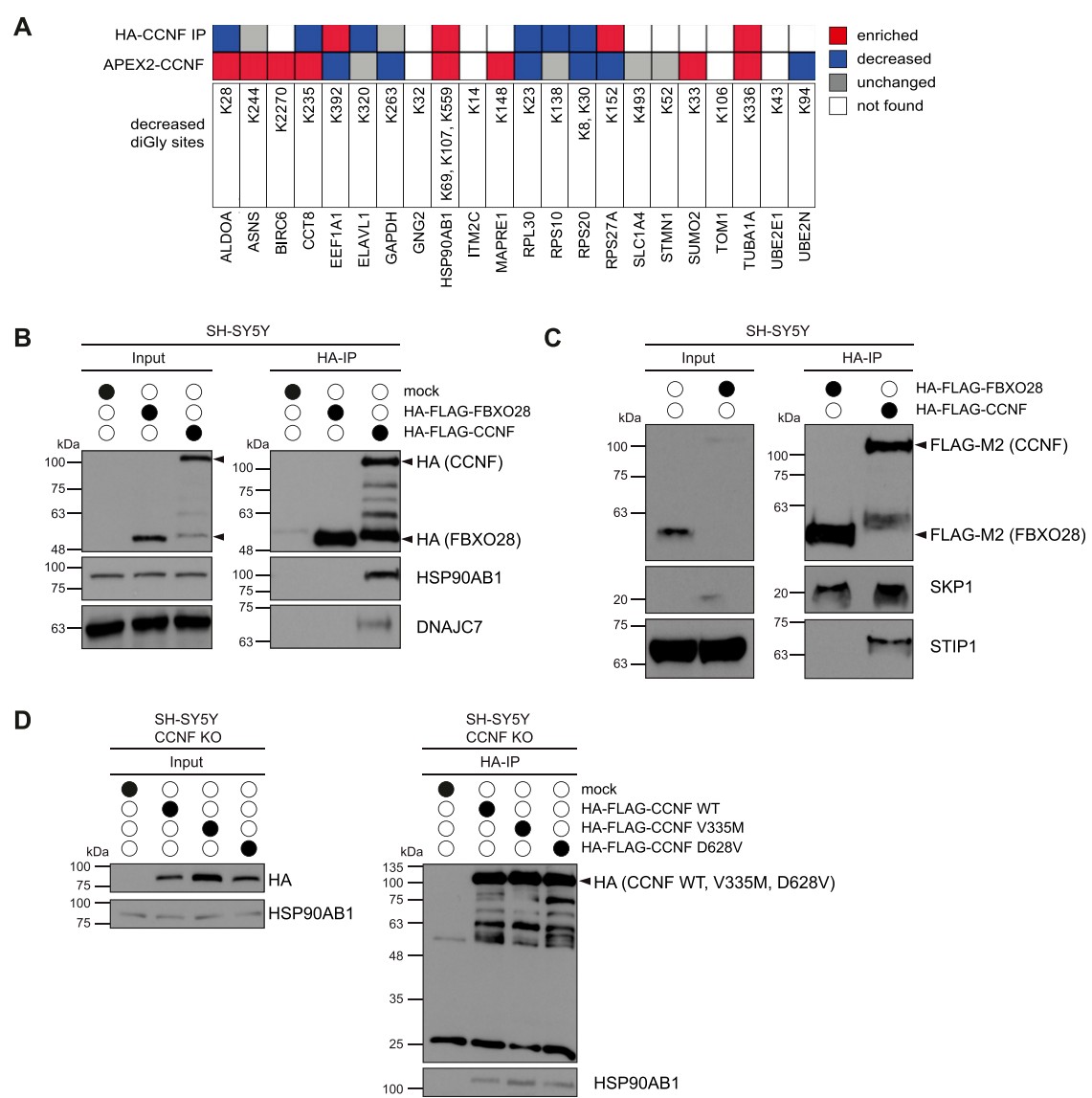

**Figure 3. Identification of HSP90AB1 as potential Cyclin-F target.**
**(A)** Overview of CCNF proximity partners and interaction candidates with decreased diGly sites in CCNF KO and mutant cells. Proteins found enriched or decreased by proximity proteomics (APEX2-CCNF) and/or immunoprecipitation mass spectrometry (HA-FLAG-CCNF IP) are marked red and blue, respectively. Unchanged proteins are marked in grey, whereas proteins that were not found are indicated with white boxes. **(B, C)** Lysates from parental (mock), HA-FLAG-FBXO28, or HA-FLAG-CCNF overexpressing SH-SY5Y cells were subjected to HA immunoprecipitation (IP), SDS–PAGE and immunoblotting. Arrows indicate specific protein bands. **(D)** CCNF KO SH-SY5Y cells re-expressing HA-FLAG-CCNF WT, V335M, or D628V or left untreated (mock) were lysed and incubated with HA-agarose followed by SDS–PAGE and immunoblotting. Arrows indicate specific protein bands.

(Fig 4A). To probe the role of cullin RING ligase (CRLs) such as SCF$^{Cyclin-F}$ in the ubiquitination of HSP90AB1, we performed denaturing IPs of HA-FLAG-HSP90AB1 from cells treated with the proteasome inhibitor Bortezomib or the NAE1 inhibitor MLN4924. Note that the latter blocks neddylation of cullins which is required for CLR activity. Intriguingly, Btz treatment led to a massive ubiquitination of HSP90AB1 which could be completely reversed by additional MLN4924 treatment (Fig 4B). Next, we asked whether HSP90AB1 ubiquitination was indeed dependent on Cyclin-F. Thereto, we used tandem ubiquitin binding entities (TUBE) with a preference for K48- and K63-linked Ub to examine the observed

decrease in abundance for a number of diGly sites on HSP90AB1 in CCNF KO cells. Lysates from SH-SY5Y CCNF WT and KO cells were subjected to pull-downs with GST-TUBE. Immunoblot analysis with specific antibodies showed decreased protein levels of HSP90AB1 in cells lacking CCNF, whereas p62 (alias SQSTM1), a known but Cyclin-F unrelated ubiquitination target, was unaffected (Fig 4C). Notably, ubiquitination of p62 was detected in our diGly proteomics but did not show any changes in CCNF KO or mutant cells. Because ubiquitination can either be a signal for degradation or exert regulatory functions on the modified protein, we monitored the abundance of HSP90AB1 in SH-SY5Y CCNF WT and KO cells grown in

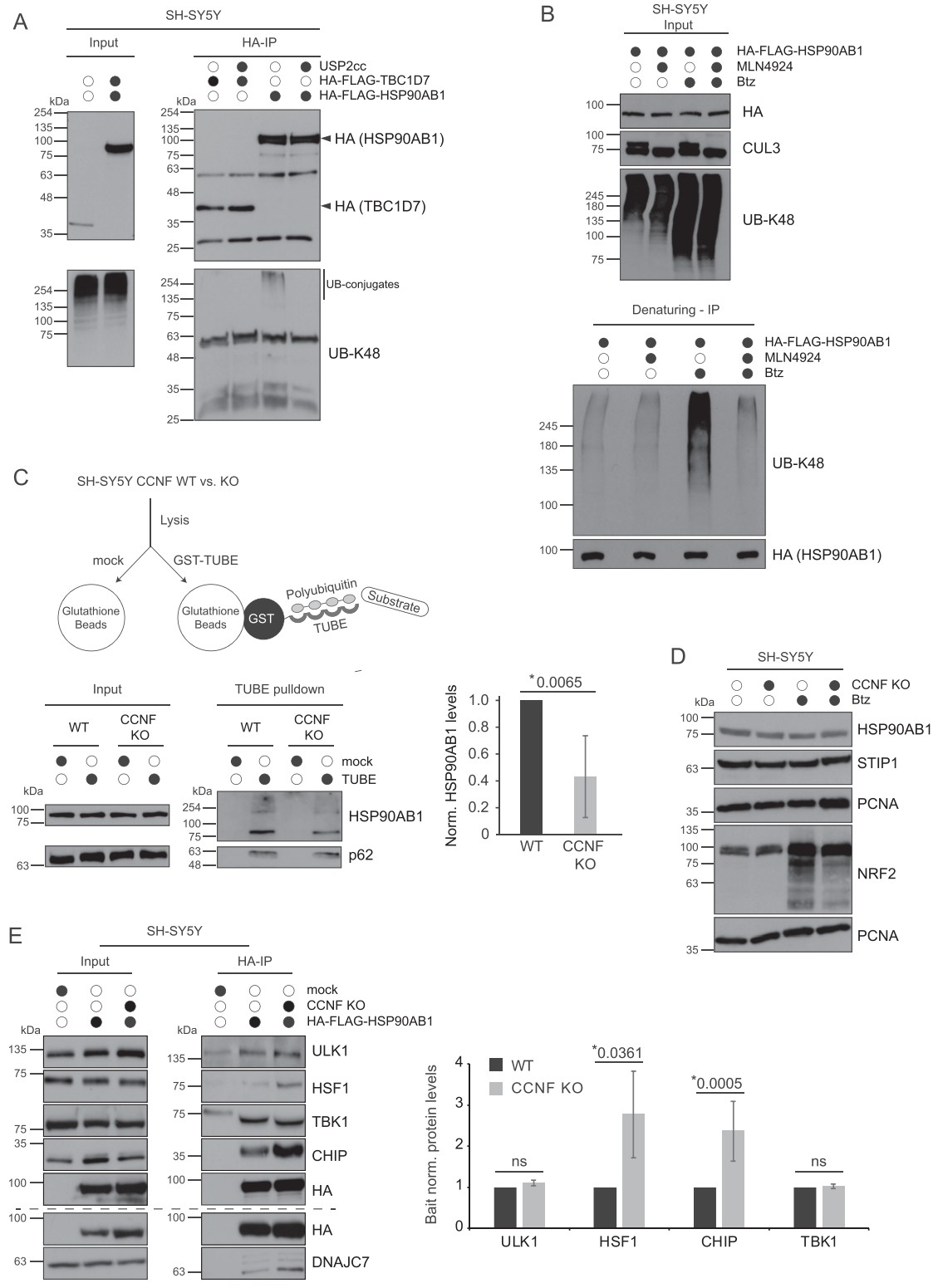

**Figure 4. Ubiquitination of HSP90AB1 by Cul1^Cyclin-F leads to altered client binding.**
**(A)** SH-SY5Y cells expressing HA-FLAG-TBC1D7 or -HSP90AB1 were lysed under denaturing conditions, differentially treated with USP2cc and subjected to HA-IP. Arrows indicate specific protein bands. **(B)** SH-SY5Y cells expressing HA-FLAG-HSP90AB1 were grown in the absence or presence of MLN4924 and/or Bortezomib (Btz) followed by denaturing HA-IP. **(C)** Schematic representation of TUBE pull-downs (upper panel). Ubiquitin conjugates in lysates from SH-SY5Y CCNF WT and KO cells were enriched using TUBE pull-downs and detected by specific antibodies (lower left panel). Bar graph shows quantification of HSP90AB1 in TUBE pull-down (lower right panel). Statistical analysis (n = 3) of HSP90AB1 levels was performed using two-sided, unpaired t test. Data represent mean ± SD. **(D)** SH-SY5Y CCNF WT and KO cells were grown in the

the absence and presence of Btz. Whereas NRF2, a protein with fast-turnover by the proteasome, increased in abundance upon Btz treatment, the levels of HSP90AB1 and of one of its co-chaperones STIP1 were unaffected by either loss of CCNF, proteasome blockage or both (Fig 4D) Similar results were obtained in cycloheximide chase assays with endogenous and overexpressed HSP90AB1 (Fig S4A). This suggests a non-degradative function for the Cyclin-F–dependent ubiquitination of HSP90AB1. The cytosolic HSP90 complex is formed by homodimers of HSP90AA1 or HSP90AB1, assists in proper folding, stabilization and activation of client proteins and is highly regulated by co-chaperones and post-translational modifications (PTMs) such as phosphorylation. To gain first mechanistic insights into the relevance of HSP90 ubiquitination for its functionality, we monitored the binding of HSP90AB1 to some of its co-factors and client proteins. To this end, we expressed HA-FLAG-HSP90AB1 in SH-SY5Y CCNF WT and KO cells and performed HA-IPs following mild lysis. Remarkably, in CCNF KO cells, we detected significantly increased HSP90AB1 binding to the HSP90 client heat shock factor 1 (HSF1) which is a key component of the proteotoxic stress response. Moreover, the HSP90 co-factors and potential CCNF ubiquitination targets DNAJC7 and CHIP also showed enhanced binding to HSP90AB1 when its ubiquitination was reduced due to the lack of CCNF (Figs 4E and S4B). The fact that ULK1, TBK1, and STIP1 did not increase in abundance in HSP90 immunoprecipitates indicates that the effect of HSP90AB1 ubiquitination on its chaperone function might be client and co-factor specific (Fig 4E). Overall, these findings suggest that Cyclin-F controls HSP90 function in a ubiquitin dependent manner.

## Discussion

In this study, we acquired ubiquitinome, proximity and interaction data to elucidate potential functions of SCF[Cyclin-F] in neuronal, respective patient-derived cells. Integration of these different data sets unveiled Cyclin-F association with cellular chaperone machinery components and HSP90AB1 ubiquitination by Cyclin-F. Remarkably, this ubiquitination is diminished in cells lacking CCNF or expressing mutated CCNF variants linked to ALS, implicating a potential loss-of-function mechanism for these mutants.

Missense mutations in CCNF contribute to the development of ALS (25, 27, 30). In this context, a number of studies proposed a gain of toxic function mechanism based on observations by semi-quantitative immunoblotting that levels of K48 linked polyUb increased upon expression of Cyclin-F S621G, K97R, or S195R (25, 26). In contrast, our mass spectrometry-based quantitative ubiquitinome profiling of ALS patient-derived LCLs carrying V335M and D628V mutant CCNF uncovered a prominent decrease in signature diGly peptides for K48- and K63-linked Ub chains. Consistent with this notion, diminished E3 ligase activity was reported for the CCNF mutation S509P (26). Moreover, several diGly sites that were found decreased in CCNF KO cells were regulated in the same direction in both ALS patient-derived LCLs. Based on these findings, we propose

that the CCNF mutations V335M and D628V contribute to ALS pathogenesis via a loss of function mechanism. Although the molecular basis of Cyclin-F V335M– and D628V–driven dysfunction requires further analysis, reduced substrate ubiquitination is unlikely caused by changes in substrate-binding affinity or faulty SCF[Cyclin-F] complex assembly because we did not observe altered HSP90AB1 binding of CCNF V335M and D628V and SCF[Cyclin-F] complex formation was shown to be unaffected for several other CCNF mutants (30). Another scenario leading to a loss of function might be an altered presentation of substrate lysines towards RBX1 bound E2~Ub. Along this line, the V335M mutant might induce a conformational change in the substrate-binding cyclin domain which could result in diminished ubiquitination. Alternatively, V335M and D628V might alter the spatial arrangement of Cyclin-F in such ways that RBX1-mediated Ub transfer to designated targets is blocked or reduced by steric hindrances.

Intriguingly, we identified HSP90AB1 as a ubiquitinated substrate of the SCF[Cyclin-F] complex under basal, housekeeping conditions. HSP90AB1 is constitutively expressed and homodimerizes to yield a functional HSP90 complex. Nonetheless, HSP90AB1 can also built heterodimers with its structurally very similar but inducible isoform HSP90AA1 (35). The protomers of the HSP90 family share a common domain structure, consisting of an N-terminal domain (NTD) linked to the middle domain (MD) by an unstructured charged-linker region and a C-terminal domain (36, 37). The NTD forms a nucleotide-binding pocket, important for ATP binding as well as co-chaperone binding, whereas the MD is not only binding co-chaperones but is also important for HSP90 client binding. The carboxy-terminus of HSP90 protomers allows the constitutive dimerization of the complex and contains the MEEVD motif, crucial for the interaction with tetratricopeptide-containing repeat domain (TPR domain) containing proteins (38, 39). These TPR domains are present in several co-chaperones, including STIP1 (37, 40). The HSP90 cycle starts in an open position that transitions to a closed conformation by ATP hydrolysis with the help of co-chaperones. Thereby client maturation is enabled. This process is highly regulated at several levels by co-chaperones, PTMs and even by client binding itself (37). Phosphorylation, for example, generally decreases HSP90 ATPase activity, affects co-chaperone binding and client dynamics (41). Other PTMs such as acetylation (e.g., of K69) has reported effects on co-chaperone binding, ATP binding and cellular functions of the HSP90 complex (41, 42, 43). Interestingly, we discovered that K69 can also serve as ubiquitin acceptor sites, suggesting equal or opposing effects on HSP90AB1 function or localization. Other diGly sites in HSP90AB1 found in this study distribute throughout the protein (K107, K180, K275, K286, K435, K438, K531, K559, K568, K574, and K607) and unlikely serve as proteasomal eat-me signal because HSP90AB1 protein levels were neither changed by Btz treatment nor by the lack of Cyclin-F–mediated ubiquitination. Our findings seem to contradict previous data showing HSP90 ubiquitination by CHIP at several sites (e.g., K107, K204, K219, K275, K284, K347, and K399) and proteasomal degradation

absence or presence of Btz followed by lysis and immunoblotting. **(E)** SH-SY5Y WT and KO cells expressing HA-FLAG-HSP90AB1 or mock were subjected to HA-IP before SDS–PAGE and immunoblot analysis. Bar graph shows quantification of ULK1, HSF1, CHIP, and TBK1 in HSP90AB1 immunoprecipitates. Statistical analysis (n = 3) of HSP90AB1-binding proteins levels was performed using two-sided, unpaired $t$ test. Data represent mean ± SD.

of HSP90AA1 in HEK cells ([41], [44], [45]). However, because HSP90AA1 is an inducible isoform, it might be regulated differently than the constitutively expressed HSP90AB1. It is noteworthy that we found some of the reported CHIP ubiquitination sites to also be potential targets of Cyclin-F (K107, K275, and K607), indicating possible redundancy between E3 ligases or different layers of Ub-dependent regulation. The increased binding of HSP90AB1 to its client HSF1 as well as to its co-factor CHIP in the absence of CCNF might provide possible explanations for likely regulatory functions of HSP90AB1 ubiquitination. Enhanced binding of clients could stem from chaperone complex stalling because of a decrease in ATPase activity or altered co-chaperone binding, leading to a change in complex dynamics and a potential blockage of the HSP90 cycle. These possibilities are in line with diGly sites found in the N-terminal part of HSPAB1 important for ATPase activity (K69 and K107) as well as with diGly sites located in domains responsible for co-chaperone binding (K69, K107, and K559).

HSP90 is involved in early embryonic development, germ cell maturation, cytoskeletal stabilization, cellular transformation, signal transduction, long-term cell adaptation and many other cellular processes through its diverse set of client proteins ([35]). Therefore, HSP90 dysfunction by altered regulatory PTMs might have detrimental effects on cells in the context of ALS. Our observation of a deregulated Cyclin-F–HSP90 axis could implicate effects on folding stress response (HSF1) and HSP90 function itself (CHIP) ([46], [47], [48], [49], [50]). These processes might converge on disturbed protein homeostasis, a typical phenomenon observed in ALS ([23]). Conversely, impairment of the proteasome as observed, for example, by C9orf72-derived dipeptide repeat proteins ([51]) might lead to a backlog within the UPS and affect the activities of E3 ligases such as SCF^Cyclin-F. While this hypothesis requires further testing, HSP90 functions might thus be generally compromised in ALS disease conditions even in the absence of mutations in CCNF.

# Materials and Methods

## CCNF ALS patient cells

The LCL line with p.D628V mutation was derived from a male patient with spinal onset of familial ALS at the age of 47 yr. Both his father and paternal grandfather were affected by the disease, in agreement with an autosomal-dominant mode of inheritance. The patient did not suffer from FTD comorbidity. Because of the loss of follow-up, the survival status of the patient is unknown. The LCL line with the p.V335M mutation was derived from a female ALS patient without a family history for the disease. She also had a spinal onset of disease at the age of 62 yr with distal extensor weakness in the lower extremities, followed by paresis in the upper extremities and subsequently bulbar symptoms. She had clinical signs of both upper and lower motor neuron degeneration. Sensory function and coordination were unremarkable. Both patients were subject to whole exome sequencing, and genetic variants in other known ALS disease genes were excluded. For the collection and use of blood cells from ALS patients as well as for whole exome sequencing of blood DNA, written informed consent was obtained from all individuals. The experiments have been approved by the local ethical committees of the Medical Faculties Ulm (Ulm University) and Mannheim (ethical committee II of the University of Heidelberg). Approval numbers are Nr. 19/12 and 2020-678N, respectively.

## Transfections and treatments

Transfections were performed with 1 $\mu$g DNA per plasmid added to 200 $\mu$l OptiMEM (Invitrogen), Lipofectamine 2000 (Invitrogen) or X-tremeGENE HP (Roche) in a 1:3 ratio ($\mu$g DNA: $\mu$l of transfection reagent) and incubated for 20 min at RT prior to addition to cells. Cells were treated with 1 $\mu$M Bortezomib (Btz) for 8 h. Lysates were incubated with USP2 at 0.625 $\mu$M for 4 h at 4°C in an overhead shaker. CHX chase was performed using 100 $\mu$g/mL of CHX for 2–8 h at 37°C. Neddylation of CRLs was inhibited with 1 $\mu$M MLN4924 for 4 h at 37°C.

**Table of reagents and resources.**

| Reagent or resource | Reference or source | Identifier or Cat. no. |
|---|---|---|
| Affinity beads/agarose | | |
| ANTI-FLAG M2 Affinity Gel | Sigma-Aldrich | A2220-1ML |
| Glutathione Sepharose 4B | GE Healthcare | 17-0756-01 |
| HA peptide | Sigma-Aldrich | I2149-1MG |
| Anti-HA agarose | Sigma-Aldrich | A2095-5X1ML |
| Pierce Anti-HA agarose | Thermo Fisher Scientific | 26182 |
| PTMScan Ubiquitin Remnant Motif (K-$\varepsilon$-GG) Kit | Cell Signaling | 5562 |
| Streptavidin agarose | Sigma-Aldrich | S1638-5ML |
| UM101: TUBE 1 | Lifesensors | UM-0101-1000 |
| Primary Antibodies | | |

| Reagent or resource | Reference or source | Identifier or Cat. no. |
|---|---|---|
| α-tubulin | Abcam | ab7291 |
| Anti-KAP1 antibody | Abcam | ab22553 |
| APEX (IgG2A) | Regina Feederle | Custom made |
| Biotin | Pierce | 31852 |
| Biotin FITC | Abcam | ab6650 |
| Cyclin-F | Santa Cruz | sc-952 |
| DNAJC7 | Proteintech | 11090-1-AP |
| FlaG M2 | Cell signaling | 2368 |
| HA.11 | Covance/BioLegend | MMS-101P/901501 |
| HSF1 | Cell Signaling | 4356 |
| HSP90β | Cell Signaling | 7411 |
| Ub K48 | Cell Signaling | 8081 |
| Nrf2 | Abcam | ab62352 |
| p62/SQSTM1 | BD | 610832 |
| PCNA | Santa Cruz | sc-7907 |
| PCNA (PC10) | Santa Cruz | sc-56 |
| Skp1 | Cell Signaling | 2156 |
| STIP1 | Abcam | ab126724 |
| STUB-1/CHIP | Bethyl Laboratories | A301-572A |
| TBK1/NAK | Abcam | ab40676 |
| Tollip | Abcam | ab187198 |
| Secondary ABs | | |
| Anti-goat-HRP | Promega | V8051, RRID: AB_430838 |
| Anti-mouse-HRP | Promega | W402B |
| Anti-rabbit-HRP | Promega | W401B |
| Anti-rat-HRP | Sigma-Aldrich | A-9037, RRID: AB_258429 |
| Software/Tools | | |
| Adobe Illustrator 2022 | Adobe | |
| Fiji, ImageJ | N/A | Version 1.53j |
| gRNA design Tool | portals.broadinstitute.org/gpp/public/analysis-tools/sgrna-design | |
| gRNA design Tool - CRISPOR | crispor.tefor.net | |
| MaxQuant | N/A | Version 1.6.0.1 |
| Perseus | N/A | Version 1.6.10.43 |
| QuikChange Primer Design | https://www.agilent.com/store/primerDesignProgram.jsp | |
| R | | Version 4.1.1 |
| R-Studio | | Version 1.4.1717 |
| Plasmids/Vectors | | |
| ORF CCNF | Dharmacon | MHS6278-202831979 |
| ORF FBXO28 | Dharmacon | OHS1770-202323126 |
| ORF HSP90AB1 | Horizon Discovery | MHS6278-202807158 |
| pHAGE-N-myc-APEX2 | Zellner et al (2021) (52) | |
| pMD2.G | Addgene | 12259 |

| Reagent or resource | Reference or source | Identifier or Cat. no. |
|---|---|---|
| psPAX2 | Addgene | 12260 |
| pSpCas9(BB)-2A-Puro (PX459) V2.0 | Addgene | 62988 |
| gRNAs/Primers | | |
| sgRNA1_mouse-Seq: CACCGAGACAACACGTATAAATACG | Thermo Fisher Scientific | Custom order |
| sgRNA2_mouse-Seq: CACCGGTAACTGACACTCCGCTCGG | Thermo Fisher Scientific | Custom order |
| sgRNA1_human CCNF- Seq: ACACCGCGTTTGGTTCTCCGCCCCGAG | Thermo Fisher Scientific | Custom order |
| sgRNA2_human_CCNF-Seq: ACACCGGTAGACCACGGTGACATCGG | Thermo Fisher Scientific | Custom order |
| Sequencing primer human CCNF gRNA1_fw TTTGTCCATGTGGTGTGTGT | Thermo Fisher Scientific | Custom order |
| Sequencing primer human CCNF gRNA1 rev TGAGATAGGAGAGGCGGGT | Thermo Fisher Scientific | Custom order |
| Sequencing primer human CCNF gRNA2 fw TTTCCCGGTTGCTTGCTT | Thermo Fisher Scientific | Custom order |
| Sequencing primer human CCNF gRNA2 rev CATGTCCTCCTCCTCACT | Thermo Fisher Scientific | Custom order |
| Sequencing primer mouse CCNF - gRNA2_fw GAGGAAGGTGGAGGATGT | Thermo Fisher Scientific | Custom order |
| Sequencing primer mouse CCNF - gRNA2_rev TCTCCTACAACTACTCCC | Thermo Fisher Scientific | Custom order |
| Sequencing primer mouse CCNF - gRNA1_fw GGGTTATGTAGGGGTCAG | Thermo Fisher Scientific | Custom order |
| Sequencing primer mouse CCNF - gRNA1_rev AGACAAGAGGGAGGAAAA | Thermo Fisher Scientific | Custom order |
| Cell lines | | |
| LCL, ctrl 1, female | Jochen Weishaupt | This study |
| LCL, ctrl 2, female | Jochen Weishaupt | This study |
| LCL, CCNF V335M, female | Jochen Weishaupt | This study |
| LCL, ctrl 3, male | Jochen Weishaupt | This study |
| LCL, ctrl 4, male | Jochen Weishaupt | This study |
| LCL, CCNF D628V, male | Jochen Weishaupt | This study |
| N2a cells | ATCC | CCL-131 |
| SH-SY5Y cells | ATCC | CRL-2266 |
| Kits | | |
| Amaxa SF Cell Line 4D-Nucleofector X Kit | Lonza | V4XC-2012 |
| Pierce BCA Protein Assay Kit | Thermo Fisher Scientific | 23225 |
| PureLink Genomic DNA Kit | Invitrogen | K1820-02 |
| QIAprep Spin Miniprep kit | QIAGEN | 27106 |
| Chemicals/enzymes | | |
| Acrylamide solution | PanReac AppliChem | A0951 |
| Ammonium bicarbonate | Sigma-Aldrich | 9830 |
| BbsI | NEB | R0539S |
| Benzonase | Merck Millipore | 71205-3 |
| Biotin-Phenol | Iris Biotech | LS-3500.5000 |
| Bortezomib 99% | LC Labs | B-1408 |

| Reagent or resource | Reference or source | Identifier or Cat. no. |
|---|---|---|
| BSA | Sigma-Aldrich | A8022-100G |
| Complete | Roche | 4693132001 |
| Disodiumhydrogenphoshpate | Merck | 1.06580.5000 |
| Dithiothreitol | Sigma-Aldrich | 43815-5G |
| Dulbecco's PBS | Thermo Fisher Scientific | 14190169 |
| Ethylenediaminetetraacetic acid | Merck | 1.008418.1000 |
| Glycerol | Roth | 3783 |
| GO-Taq polymerase | Promega | M784B |
| Hydrogenperoxide | Sigma-Aldrich | H1009 |
| IGEPAL CA-630 (NP-40) | Sigma-Aldrich | I8896 |
| KOD Hot Start DNA Polymerase | Sigma-Aldrich | 71086 |
| Lipofectamine 2000 | Invitrogen | 11668-019 |
| N-Ethylmaleimide (NEM) | Sigma-Aldrich | E3876-5G |
| Opti-MEM | Invitrogen | 31985-062 |
| Paraformaldehyde solution 4% | Chemcruz | sc-281692 |
| Phenylmethylsulfonyl fluoride | Sigma-Aldrich | P7626-1G |
| PhosSTOP | Roche | 4906837001 |
| ProLong Gold Antifade | Invitrogen | P36931 |
| Puromycin dihydrochloride | Sigma-Aldrich | P8833-100mg |
| Pwo-polymerase | VWR | 01-5010-88 |
| Recombinant USP2 | R&D Systems | E-504-050 |
| Sodium L-ascorbate | Sigma-Aldrich | A7631 |
| TCEP | ROTH | HN95.2 |
| Trifluoroacetic acid | Honeywell Fluka | 302031-100ML |
| Triton X-100 | Merck | 1.08603.1000 |
| Trolox | Sigma-Aldrich | 238813 |
| Tropix I-Block | Appliedbiosystems | T2015 |
| Trypsin, sequencing-grade | Promega | V5113 |
| Western Lightning Plus-ECL | PerkinElmer | NEL104001EA |
| X-tremeGENE HP | Roche | 06 366 236 001 |
| Hardware/Consumables | | |
| 4D-NucleofectorTM X Unit | Lonza | |
| Amersham Protran 0.45 $\mu$m NC | GE Healthcare Life Science | 10600002 |
| EMPORE Octadecyl C18 47 mm | Supelco Analytical | 66883-U |
| Lyophilisator, Alpha 1-2 LD Plus | CHRIST | N/A |
| Mini-PROTEAN Tetra cell | Bio-Rad | 1658004EDU |
| Mini-Transblot cell | Bio-Rad | 1703930 |
| Sonifier | SONIFIER Branson | W-250D |
| Super RX-N | Fujifilm | 47410 19289 |
| Ultrafree-MC, HV 0.45 $\mu$m | Merck Millipore | UFC30HV00 |
| Vacuum Centrifuge | Eppendorf | N/A |
| Zeiss LSM800 oil 60× objective | Zeiss | N/A |
| Easy-NLC1200 | Thermo Fisher Scientific | N/A |
| QExactive[HF] mass spectrometer | Thermo Fisher Scientific | N/A |

## Plasmid and cell line generation

PCR was performed to add attB sites to ORFs and cloned into pDONR223. Using recombinational cloning these ORFs were then moved to the following destination vectors: pHAGE-N-Flag-HA, pHAGE-C-FLAG-HA, and pHAGE-N-myc-APEX2 (52). Stable cell lines in SH-SY5Y and N2a cells were generated by lentiviral transduction. 1 µg pMD2.G, 1 µg pPAX2, and 1 µg destination vector was used for transfection. Puromycin (2 µg/ml) was added to the cells 24 h post transduction for selection.

## Mutagenesis

Mutagenesis primers were designed using QuikChange Primer Design software (Agilent Technologies). KOD Hot Start or Pwo DNA Polymerase (Merck Millipore) were used according to the manufacturers' protocols. For Pwo DNA polymerase elongation periods were extend to 14 min per cycle. The PCR product was purified with the QIAquick PCR purification kit (QIAGEN) and amplified in *Escherichia coli*.

## Knockout generation and validation

First single guide RNAs (sgRNAs) were designed using the sgRNA-design tool of the Broad Institute, CRISPick or CRISPOR (crispor.tefor.net) (53, 54 *Preprint*, 55). sgRNAs were provided with sticky end overhang sequences for ligation with BbsI-digested pSpCas9(BB)-2A-Puro V2.0 vector which was a gift from Feng Zhang (Addgene plasmid # 62988; http://n2t.net/addgene:62988; RRID:Addgene_62988) (56). After ligation cells were either transfected with two sgRNAs using XtremeGene HP DNA transfection reagent according to manufacturer's instructions (for N2a) or by electroporation with the 4D-Nucleofector X Unit using the Amaxa SF Cell Line 4D-Nucleofector X Kit according to manufacturer's instructions (for SH-SY5Y). 24 h post-transfection cells were grown in 4 µg/ml Puromycin for 48 h. Single cells were selected using serial dilution. Genomic DNA was purified (Invitrogen) and amplified by touchdown PCR using the GO-taq polymerase (Promega). Proper genome editing was verified by Sanger sequencing (by Eurofins Genomics EU) and immunoblotting.

## Immunofluorescence

Cells were seeded on coverslips and washed three times with DPBS (GIBCO) followed by fixation with 4% PFA (Santa Cruz) for 10 min and permeabilization with 0.5% Triton X for 10 min. Subsequently, cells were blocked with 1% BSA in PBS for 1 h at RT. Fluorophore-coupled primary antibodies were incubated for 1 h at RT in the dark. Coverslips were mounted with mounting solution (Prolonged Gold with DAPI; Invitrogen) on microscope slides (Thermo Fisher Scientific) and imaged using a confocal microscope Zeiss LSM800 with a 63× magnification oil-immersion objective. Image analysis was performed with ImageJ 1.53j (Fiji).

## Immunoblotting

Cells were washed with DPBS before harvesting by scraping on ice. Lysis was performed with RIPA buffer (50 mM Tris, pH 7.5, 150 mM NaCl, 0.1% SDS, 0.5% sodium desoxycholate, 1% Triton X, PhosStop [Roche], and protease inhibitor [Roche]) and protein concentrations were adjusted using BCA assays. Protein samples were separated by SDS–PAGE (100V) and transferred (2 h 15 min, 0.3 A) on a nitrocellulose membrane (0.45 µm pore size). Membranes were blocked with I-Block (Invitrogen) and incubated at 4°C overnight with the primary antibody. After washing three times with TBS-T-buffer, secondary antibody coupled to horseradish peroxidase was added to the membrane for 1 h at RT. Immunoblots were washed another three times with TBS-T before enhanced chemiluminescence analysis using ECL (PerkinElmer) and x-ray films (Fuji Medical).

## Immunoprecipitation

Cells grown in 2–4 × 15 cm cell culture plates per sample were harvested by scraping on ice and stored at –80. Lysis was performed for 30 min at 4°C with MCLB buffer (50 mM Tris HCl, pH 7.5, 150 mM NaCl, 0.5% NP40, 1× PhosStop, and 1× protease inhibitor) or glycerol buffer (20 mM Tris, pH 7.5, 150 mM NaCl, 10% glycerol, 5 mM EDTA, 0.5% Triton X, 1× PhosStop, inhibitor, and 1× protease inhibitor). Samples were cleared from debris by centrifugation (20,000$g$ for 10 min at 4°C) and Ultrafree-CL spin-filter tubes (Millipore CL 0.45). Protein concentrations of lysates were adjusted following determination by BCA and samples incubated overnight with pre-equilibrated anti-HA-agarose (Sigma/Pierce Anti-HA Agarose) or anti-Flag M2 affinity gel (Merck Millipore) using an overhead shaker at 4°C. Subsequently, agarose beads were washed five times with the respective buffer and eluted by boiling in SDS sample buffer (200 mM Tris-HCL, 6% SDS, 20% glycerol, 300 mM DTT, and bromophenol blue) (5 min at 95°C) or washed five more times with DPBS (GIBCO) before elution with HA peptide (Sigma-Aldrich). Eluted immune complexes were precipitated with TCA (final concentration 20%) and washed with ice cold acetone. Samples were resuspended in 50 mM ammonium bicarbonate buffer containing 10% acetonitrile and trypsinized for 4 h at 37°C.

## APEX2-mediated biotinylation

Cells were grown in the presence of 500 µM biotin-phenol (Iris Biotech) for 30 min at 37°C and pulsed with 1 mM $H_2O_2$ at RT. Biotinylation was stopped by washing three times with quencher solution (10 mM sodium azide, 10 mM sodium ascorbate, 5 mM 6-Hydroxy-2,5,7,8-tetramethylchroman2-carboxylic acid [TROLOX], DPBS). The third quenching step was performed for 15 min before washing three times with DPBS. Cells were either lysed in RIPA buffer or frozen at –80°C after adjusting cell numbers.

## Streptavidin pulldown

Biotinylated samples were thawed on ice and lysed in qRIPA buffer (50 mM Tris pH 7.5, 150 mM NaCl, 0.1% SDS, 0.5% sodium desoxycholate, 1% Triton X, 10 mM sodium azide, 10 mM sodium ascorbate, 5 mM TROLOX, PhosStop [Roche], and protease inhibitor [Roche]) for 40 min at 4°C. Samples were cleared from debris (20,000$g$, 10 min, 4°C) and incubated with pre-equilibrated streptavidin agarose beads (Sigma-Aldrich) overnight at 4°C in an overhead shaker. Pull-

downs were washed twice with RIPA buffer followed by four times washing with 3 M urea wash buffer (50 mM ABC buffer, 3 M urea). TCEP was added to a final concentration of 5 mM and incubated for 30 min at 55°C. Once cooled to RT samples were incubated with iodoacetamide (IAA) for 20 min at RT in the dark followed by addition of DTT (20 mM). Samples were then washed with 2 M urea wash buffer (50 mM ABC buffer, 2 M urea) before trypsinization overnight at 37°C.

## Mass spectrometry

Digests were stopped by the addition of formic acid and samples were desalted on custom-made stage-tips (C18 material–Supelco Analytical) (57). Using an Easy-nLC1200 liquid chromatography (Thermo Fisher Scientific), peptides were loaded onto custom filled C18 reversed-phase columns and separated using a gradient of 5%–33% acetonitrile in 0.5% acetic acid over 90 min and detected on an Q Exactive HF mass spectrometer (Thermo Fisher Scientific). Dynamic exclusion was enabled for 30 s and singly charged species or species for which a charge could not be assigned were rejected. MS data were processed and analyzed using MaxQuant (1.6.0.1) (58, 59) and Perseus (1.6.10.43). Proximity proteomics was performed in triplicates and interaction proteomics experiments were performed in quadruplicates. Unique and razor peptides were used for quantification. Matches to common contaminants, reverse identifications and identifications based only on site-specific modifications were removed before further analysis. $Log_2$ H:L ratios were calculated. $t$ tests were used to determine statistical significance between conditions. A q-value < 0.05 was considered statistically significant. $Log_2$ fold change (H:L) > 2 and >1 was used as cutoff for experiments involving HA-IPs and APEX2, respectively. Functional annotation enrichment analysis was performed using DAVID (60, 61) coupled to significance determination using Fisher's exact test and correction for multiple hypothesis testing by the Benjamini and Hochberg FDR.

## diGly proteomics

Cells were cultured in lysine- and arginine-free DMEM supplemented with dialyzed FBS, 2 mM L-glutamine, 1 mM sodium pyruvate, penicillin/streptomycin, and light (K0) lysine (38 $\mu$g/mL) and arginine (66 $\mu$g/ml). Heavy medium was the same except the light lysine was replaced with K8-lysine (L-Lysine, 2HCl U-13C U-15N, Cambridge Isotope Laboratories Inc). Cells were processed as essentially as described in Fishkin et al (2016) (62). Briefly, cells were washed twice with ice-cold PBS and lysed in 5 ml denaturing lysis buffer (8M urea, 50 mM Tris [pH 8], 50 mM NaCl, 1× PIC [protease inhibitor cocktail, EDTA-free; Roche], 50 $\mu$M DUB inhibitor PR-619 [Millipore]). Samples were incubated on ice for 10 min and then sonicated with 3 × 20 s pulses. After removal of non-solubilized material (15,000$g$/10 min), differentially labeled lysates were mixed at equal ratios based on total protein determined by BCA (Pierce-Thermo; typically, 10 mg of total protein). After reduction with 5 mM DTT and alkylation with 10 mM chloroacetamide, lysates were digested with 5 ng/$\mu$l lys-C (Wako) for 1 h at RT. Subsequent digestion of peptides with trypsin (Promega) was performed as described (63). Lyophilized peptides were resuspended in 1.5 ml IAP

buffer (50 mM MOPS [pH 7.4], 10 mM $Na_2HPO_4$, and 50 mM NaCl) and centrifuged to remove any insoluble material (2,500$g$/5 min). The supernatant was incubated with anti-diGly antibody (32 $\mu$g/IP) conjugated to protein A agarose beads (Cell Signaling) for 1 h at 4°C. Unbound peptides were removed through 3× washing with IAP buffer and once with PBS. Bound material was eluted 4× with 50 $\mu$l 0.15% TFA and peptides were desalted using C18 stage-tip method (57). Each sample was immunoprecipitated sequentially two times and each IP was analyzed separately by mass spectrometry. Peptides samples were separated on a nanoflow HPLC system (Thermo Fisher Scientific) using a 226 min gradient of 5–33% acetonitrile containing 0.5% acetic acid on custom filled C18 reversed-phase columns and analyzed on a Q Exactive HF mass spectrometer (Thermo Fisher Scientific) using data-dependent acquisition selecting the most intense peaks from each full MS scan acquired in the Orbitrap for subsequent MS/MS while excluding peptides with unassigned charge states or charge states below +3 from fragmentation (see RAW files for specific settings). Raw data files from quadruplicate samples were processed with MaxQuant (1.6.0.1) as described previously (58, 59) using a human (UP000005640) UNIPROT database and the following parameter settings: first search peptide mass tolerance 20 ppm, main search peptide mass tolerance 0.5 D, tryptic digestion allowing up to two missed cleavages, cysteine carbamidomethylation (57.021464) as fixed modification, methionine oxidation (15.994946), N-terminal protein acetylation (42.010565) and diGG (114.042927; excluded from the C terminus) as variable modifications, revert decoy mode and peptide, protein and site FDR ≤ 0.01. Perseus (1.6.10.43) was used for data sorting. $Log_2$ H:L ratios were calculated. $t$ tests were used to determine statistical significance between conditions. A q-value < 0.05 was considered statistically significant. $Log_2$ fold change (H:L) > 0.5 and < −0.5 was used as cutoff. Heat maps were generated using MultiExperiment Viewer (64).

## Denaturing immunoprecipitation

Cells in 2 × 15 cm cell culture dishes were harvested by scraping, washed with DPBS and frozen at −20°C. Lysis was carried out with a denaturing buffer containing 1% SDS (50 mM Tris pH 8.0, 150 mM NaCl, 0.5 mM DTT, 0.5 mM PMSF, 10 mM NEM, 0.5% NP40, 1% SDS, protease inhibitor [1×], benzonase) for 15 min at 4°C-Lysates were diluted to 0.1% SDS with NP40-buffer (50 mM Tris pH 8.0, 150 mM NaCl, 10 mM NEM, 0.5% NP40, 1× protease inhibitor) and sonicated (8 × 1 s pulse, 1 s rest) on ice. Debris was cleared by centrifugation (>20,000$g$, 10 min, 4°C) and protein concentrations adjusted across samples following BCA. Samples were incubated with pre-equilibrated anti-HA-agarose overnight at 4°C in an overhead shaker. Before elution, samples were washed either four times with USP2 wash buffer (50 mM Tris HCl, pH 8.0, 300 mM NaCl, 0.5% NP40, and protease inhibitor [1×]) and once with USP2 reaction buffer (50 mM Tris HCl, pH 8.0, 10 mM NaCl, 0.5 mM DTT, and 0.01% NP40) or five times with NP40 buffer followed by differential treatment with recombinant USP2.

## TUBE pull-down

Cells in 2 × 15-cm cell culture dishes were harvested by scraping and washed with DPBS. Cell numbers were adjusted using a cell counter (Invitrogen Countess) and samples frozen at −20°C. Cell pellets

were lysed with TUBE buffer (50 mM Tris, pH 7.5, 150 mM NaCl, 1 mM EDTA, 1% NP40, 10% glycerol, 50 $\mu$M PR-619, 5 mM 1,10-phenanthroline, 2 mM PMSF, 10 $\mu$M bortezomib, and 1× protease inhibitor cocktail) with or without 200 $\mu$g/$\mu$l GST-TUBE (Lifesensors). Samples were rotated on an overhead shaker for 10–15 min at 4°C before incubation with TBS-T (20 mM Tris, pH 7.5, 150 mM NaCl, and 0.1% Tween-20) pre-equilibrated glutathione sepharose (GE Healthcare) for 4 h on an overhead shaker at 4°C. Samples were washed four times with TBS-T and eluted with three times SDS sample buffer (200 mM Tris–HCL, 6% SDS, 20% glycerol, 300 mM DTT, and Bromophenol Blue).

## Data Availability

The mass spectrometry proteomics data have been deposited to the ProteomeXchange Consortium via the PRIDE partner repository with the dataset identifier PXD030729.

## Supplementary Information

## Acknowledgements

We thank Georg Werner for his extensive advice while creating the CCNF KO lines and we are very thankful to all members of the Behrends, Edbauer, and Haass lab for readily sharing reagents, advice, and critical discussions. This work was supported by the Deutsche Forschungsgemeinschaft (DFG, German Research Foundation) within the frameworks of the Munich Cluster for Systems Neurology (EXC 2145 SyNergy–ID 390857198) and the Collaborative Research Center 1177 (ID 259130777).

### Author Contributions

A Siebert: conceptualization, data curation, formal analysis, validation, investigation, visualization, methodology, and writing—original draft, review, and editing.
V Gattringer: investigation and methodology.
JH Weishaupt: resources.
C Behrends: conceptualization, resources, formal analysis, supervision, funding acquisition, visualization, project administration, and writing—original draft, review, and editing.

### Conflict of Interest Statement

The authors declare that they have no conflict of interest.

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
