## [Reviewer comments · Life Science Alliance]

ALS-linked loss of Cyclin-F function affects HSP90

Alexander Siebert, Vanessa Gattringer, Jochen Weishaupt, and Christian Behrends

DOI: <https://doi.org/10.26508/lsa.202101359>

Corresponding author(s): *Christian Behrends, Ludwig-Maximilians-University Munich*

Review Timeline:

Submission Date:	2021-12-30
Editorial Decision:	2022-03-03
Revision Received:	2022-07-15
Editorial Decision:	2022-08-09
Revision Received:	2022-09-02
Accepted:	2022-09-05

Scientific Editor: Novella Guidi

Transaction Report:

March 3, 2022

Re: Life Science Alliance manuscript #LSA-2021-01359

Prof. Christian Behrends
Ludwig-Maximilians-University Munich
Munich Cluster for Systems Neurology
Feodor-Lynen Strasse 17
Munich, Bayern 81377
Germany

Dear Dr. Behrends,

Thank you for submitting your manuscript entitled "ALS-relevant loss of Cyclin F function affects regulatory HSP90 ubiquitination" to Life Science Alliance. The manuscript was assessed by expert reviewers, whose comments are appended to this letter. As you will note from the reviewers comments below, the reviewers are quite positive about these findings, but do think that additional data and controls are required to support the conclusions of the study. Specifically both Rev 1 and 2 raise a similar concern and require to repeat a blot because ubiquitination difference for HSP90AB1 in fig 4B is not that convincing and validate the K69 ubiquitin antibody, showing that the antibody does not detect a band in cell expressing a K69A mutation. All the other points should be addressed as well. We, thus, encourage you to submit a revised version of the manuscript back to LSA that responds to all the reviewers' points.

Thank you for this interesting contribution to Life Science Alliance. We are looking forward to receiving your revised manuscript.

Sincerely,

B. MANUSCRIPT ORGANIZATION AND FORMATTING:

Reviewer #1 (Comments to the Authors (Required)):

LSA-2021-01359

ALS-relevant loss of Cyclin F function affects regulatory 1 HSP90 ubiquitination

Siebert et al

Summary: The SCF family of cullin ring ligases play critical roles in normal cell physiology and many are implicated in disease progression. SCF ligases use interchangeable substrate receptor F-box proteins to designate substrates for ubiquitination and degradation. Cyclin F is the founding member of the F-box family and plays important roles in cell cycle. Recently, mutations in cyclin F were identified in patients with ALS, although it remains unclear how these mutations contribute to disease. Determining these mechanisms will shed light on disease pathogenesis. The authors here undertake a comprehensive analysis of proteome wide changes, using both cyclin F KO cell lines and those reconstituted with mutant proteins based on those observed in ALS patients. The combine this with traditional APMS, proximity labeling MS and ubiquitin proteomics. Together, these data provide a comprehensive snapshot of cyclin f regulated proteomes. Based on the recommendations below, I am supportive of its publication.

Major points.

- 1- It was not clear if their analyses recovered any of the known cyclin F substrates, several of which have been identified in the last several years. The authors should mention how at least some scored in their MS experiments, and if they are not appearing as upregulated or as interactors, why that might be the case in their system (different cell lines, differences in cell cycles, etc.)
- 2- They go on to show new cyclin F interactors, some of which might be regulated by Cyclin F dependent ubiquitination. However, it is unclear that this interaction is relevant to ALS, since binding and ubiquitination appear unaffected by cyclin F mutations. I therefore think that the title and abstract should be scaled back, as they give the strong impression that these mutations and the interaction with HSP are relevant in disease context.
- 3- The major strength of this manuscript lies in its comprehensive, detailed and quantitative MS analysis. I therefore am supportive of publication assuming these datasets are made available, which was not clear from the current submission.
- 4- The ubiquitination difference for HSP90AB1 in 4B is not that convincing. The blot is "smudgy" and the difference appears minimal. This should be addressed with a better blot and would be supported by cyclin F overexpression. Along those same lines, the K69 ubiquitin antibody is not validated and is therefore not convincing. Showing that the antibody does not detect a band in cell expressing a K69A mutation, would strengthen this reagents confidence. It is unclear why this antibody would detect a single band and not a smear of polyubiquitination, as might be suggested by 4B.
- 5- Even on long exposure, there appears to be more HSP90AB1 in the input and pulldown of CCNF KO cells in 4F. That is a very interesting finding, but interpretation is complicated by this discrepancy, since increased binding to other factors could be due to this difference. Can this be repeated in conditions where those differences are normalized?
- 6- How many times were individual experiments in figures 3 and 4 performed?

Reviewer #2 (Comments to the Authors (Required)):

In the manuscript entitled "ALS-relevant loss of Cyclin F function affects regulatory HSP90 ubiquitination," Siebert and colleagues take a wide systems-level approach to determine possible SCF-cyclin F substrates and interacting partners. They perform most of their experiments as a compare and contrast with wild type cyclin F and mutants of cyclin F that have been linked, genetically, to ALS. They perform a global analysis of ubiquitination site changes in neural-like cell models lacking cyclin F and ALS patient-derived lymphoblastoid cell lines with two separate cyclin F mutations. The authors then go on to perform both traditional affinity based and APEX-based proximity interaction proteomics analysis of wild type cyclin F. From these global analyses, the authors focus on the interaction with cyclin F and HSP90AB1. The authors attempt to demonstrate a specific

interaction with cyclin F and HSP90AB1 as well as document that ubiquitination of HSP90AB1 is altered in cells lacking cyclin F. Lastly, they document a modest change in HSP90AB1 association with HSF1 and CHIP/STUB1 in cyclin F knockout cells. Combined, the studies are interesting, and the larger datasets will be useful additions to researchers interested in cyclin F regulation. However, some critical controls are missing and the authors, in the end, identify two interactions with highly abundant proteins (HSP90 and tubulin) as being specific and possibly informing on cyclin F function with regards to contributions to ALS-phenotypes. However, additional data supporting these interactions as being specific or regulatory (as the title suggests) is needed. Further, additional data is needed to state that cyclin F-mediated ubiquitination of HSP90 regulates HSP90 function. As currently constructed, the data in the manuscript does not support the stated results, nor do they support the conclusion described in the title of the manuscript. I would recommend adding additional data addressing the concerns detailed below or substantially altering the stated conclusions.

Major concerns:

- 1) The authors need to provide supplemental tables documenting all proteomics data (maxquant outputs) for the ubiquitinome and interaction analyses. These tables should list all identified proteins/sites for all replicates and secondary tables listing the proteins that were selected as specific and of interest based on their criteria. These tables are crucial for others to evaluate the primary data and to supplement their own studies without the need to replicate the experiments.
- 2) Did the authors perform a background proteomics analysis of total proteome changes in their four cyclin F knockout or mutant cell lines? These data are necessary to understand if changes in ubiquitination of specific proteins (up or down) correlate with changes to total protein abundance. It is understood that some of the proteins sampled by the ubiquitin-site enrichment approach may not be observed in a top-level analysis (4000 proteins) of proteome changes. However, for highly abundant proteins like Map1B, ENO1, and ribosomal proteins, their total levels could be easily assessed using already generated samples for ubiquitinome analysis.
- 3) In the end the authors identify 30 overlapping sites comparing their cyclin F KO to mutant lines. Is this overlap significant? What is expected by chance given the number of sites that are observed in all datasets? A similar analysis is needed for the overlapping interacting proteins depicted in 2E.
- 4) I appreciate that the authors utilized a control protein (TBC1D7) for their affinity-interaction proteomics to compare against cyclin F. However, the blots that show expression levels of the two proteins (figure 2A) are not done together so it is impossible to judge the relative expression levels of the two proteins (is cyclin F expression 20x as much as TBC1D7?). An HA immunoblot with both cell lines on the same blot is required to make this comparison. Differences in expression levels can have dramatic effects on identified interacting proteins (like chaperones). Indeed, there appears to be a large skew towards cyclin F interacting proteins in figure 2D (SH-SY5Y).
- 5) Despite having a control protein for the affinity interaction studies, the authors lack appropriate controls for their APEX-based proximity studies. They simply compare with and without biotin phenol, which is a fine negative background control, but a positive specificity control is needed (like APEX-TBC1D7). Again, the authors identify many putative interactors in their APEX study (impossible to judge exactly how many as the datasets are not provided) many of which (like UBAP2L) are likely non-specific and would be identified across control samples with APEX-GFP, or other appropriate controls.
- 6) The authors focus on the interaction with cyclin F and HSP90AB1 and nicely show specific binding (compared to FBXO28). This data is as consistent with cyclin F being a folding client of HSP90AB1 as opposed to the authors suggestion that cyclin F is regulating HSP90 function. The authors need to state this as a possible likely scenario (given cyclin F overexpression).
- 7) The data describing changes in HSP90 ubiquitination in response to changes in cyclin F is not convincing and more data would have to be collected/shown to make any conclusive statements regarding SCF-cyclin F dependent ubiquitination of HSP90. The data shown in figure 4A could be due to any ligase ubiquitinating overexpressed HSP90AB1. Do endogenous IPs of HSP90 show similar levels of higher-order ubiquitination? Further the TUBE enrichments in cyclin F KO and controls do not reveal convincing changes in apparent HSP90AB1 ubiquitination. These experiments would need several replicates to make any argument that the subtle changes observed are meaningful and reproducible. Do they observe the same changes in cyclin F N2A KO cells or the LCL cyclin mutant cells? The authors highlight many ubiquitination sites on HSP90AB1 in figure 4C. How many of these sites have been observed in other diGLY-based ubiquitinome studies (of which there are many now)? Do these sites increase or decrease in the presence of proteasome inhibitors? Do they increase or decrease in response to Nedd8 E1 inhibition (very relevant for a possible cyclin-F target)? The authors could easily mine these datasets to strength their argument that these HSP90 ubiquitination events may be regulatory in nature. Amazingly, the authors suggest that they have a HSP90_K69 ubiquitin site specific antibody and show that this immunoreactivity is reduced in cyclin F knock out cells. However, no validation data showing this antibody is indeed specific to K69 ubiquitinated HSP90 is provided (maybe this was done in another study that could be referenced here). Without this validation data, how is the reader to know that this antibody is specific. This is particularly important because there is a paucity of ub-site specific antibodies, and this would be a noteworthy finding.

Minor concerns:

- 1) In the figure legend for figure 1 and in the text describing the results the author say, "Using statistical analysis (one-sample t-

test with a p-value of < 0.05), we identified between 38 and 100 diGly sites (27-83 proteins) with \log_2 (SILAC) ratios < -0.5 in SH-SY5Y and LCLs as well as \log_2 (SILAC) ratios < -0.75 in N2a cells." It is clear that a statistical analysis was done to determine possible differentially regulated sites but those cutoffs aren't discussed in the legend as only \log_2 fold change cutoffs are described. Are the sites depicted in 1G only based on \log_2 fold change cutoffs or are these also below a p-value cutoff. Further, it does not appear that the authors have utilized any adjustment for p-values based on multiple hypothesis testing which is critical to do for these types of datasets. I apologize if this was done and I missed it in the methods section.

2) The authors describe some characterization studies on the S621G cyclin F mutant but then never use that same mutant in their studies. Have similar studies been done on the D628V or V335M mutants? Because the authors never use the S621G mutant, why spend so much time describing previous experiments with that mutant?

Reviewer #3 (Comments to the Authors (Required)):

Siebert et al. aim to elucidate the role of CCNF mutations in amyotrophic lateral sclerosis 1) in the substrate adaptor function of cyclin F and 2) if alterations in SCFcyclin F mediated ubiquitination are of pathophysiological relevance in patients with CCNF mutations. To date, several reports characterizing the consequences of the S621G CCNF mutation in ALS have been published, however ubiquitination targets and pathophysiological effects of other CCNF mutations in ALS still remain unclear.

Main points:

The authors focus on uncovering ubiquitination targets of cyclin F in two CCNF knockout cells as well as patient-derived lymphoblastoid cell lines. In Fig1A CCNF blots consistently show double bands. However, in SH-SY5Y and the LCL cells the upper band is marked as CCNF, whereas the N2a WT cells seem to only show the lower <100 kDa band, which is lost upon KO of CCNF, whereas the upper band does not appear in neither WT nor KO cells. Are there different isoforms of CCNF in respective cell lines? How do the authors explain this difference?

What was the reasoning behind going forward with the SH-SY5Y cell line regarding HSP90AB1 binding and ubiquitination and not N2a cells and especially mutated LCLs vs. controls?

And similarly, did the authors proceed to perform ubiquitination analyses of HSP90AB1 only with SH-SY5Y CCNF WT and KO cells or also V335M and D628V mutations, if those mutations were of special interest in this particular study?

Minor points:

There is no mention of Fig S1 in the text.

Please provide information regarding ethics approval for patient-derived materials to the methods section.

We would like to thank all the reviewers for their time, effort and their valuable comments. We greatly appreciate the reviewers' feedback and are confident that the changes we made substantially improved our paper.

Reviewer #1:

Summary: The SCF family of cullin ring ligases play critical roles in normal cell physiology and many are implicated in disease progression. SCF ligases use interchangeable substrate receptor F-box proteins to designate substrates for ubiquitination and degradation. Cyclin F is the founding member of the F-box family and plays important roles in cell cycle. Recently, mutations in cyclin F were identified in patients with ALS, although it remains unclear how these mutations contribute to disease. Determining these mechanisms will shed light on disease pathogenesis. The authors here undertake a comprehensive analysis of proteome wide changes, using both cyclin F KO cell lines and those reconstituted with mutant proteins based on those observed in ALS patients. The combine this with traditional APMS, proximity labeling MS and ubiquitin proteomics. Together, these data provide a comprehensive snapshot of cyclin f regulated proteomes. Based on the recommendations below, I am supportive of its publication.

Major points.

1- It was not clear if their analyses recovered any of the known cyclin F substrates, several of which have been identified in the last several years. The authors should mention how at least some scored in their MS experiments, and if they are not appearing as upregulated or as interactors, why that might be the case in their system (different cell lines, differences in cell cycles, etc.)

We apologize that we have not systematically included this information in the representation of our interaction and proximity proteomics results. Known interactors have now been labeled in the HA-IP and APEX2 data sets in the revised Figure 2D and the new Figure S2A. In addition, we added a heat map summary of known interactors across the different data sets as new Figure S2C. Furthermore, we added one sentence that highlights commonly detected known Cyclin F interactors (Line 191: "Among them were the SCF^{Cyclin-F} ligase components CUL1, SKP1 and RBX1 as well as a number of their targets such as CCP110, RRM2, CDC6 and SFPQ.") as well as a second sentence that points to the differential detection of known Cyclin F binding partners across datasets (Line 202: "The fact that we used different cell types and varying experimental conditions might explain the observed discrepancies in the detection and scoring of established Cyclin-F binding partners (Fig S2C).").

2- They go on to show new cyclin F interactors, some of which might be regulated by Cyclin F dependent ubiquitination. However, it is unclear that this interaction is relevant to ALS, since binding and ubiquitination appear unaffected by cyclin F mutations. I therefore think that the title and abstract should be scaled back, as they give the strong impression that these mutations and the interaction with HSP are relevant in disease context.

While HSP90AB1 binding to Cyclin-F is indeed not affected by the ALS-linked mutations V335M and D628V (as shown in Figure 3D), ubiquitination of HSP90AB1 is clearly reduced in patient derived cells carrying these mutations (as shown in Figure 1G). The fact that these two Cyclin-F mutations and Cyclin-F knockouts (in two different cell types) yielded the same

results in regard to reduced HSP90AB ubiquitination led to our conclusion that this phenotype is relevant in an ALS context. However, to avoid overstating our findings we revised the title (line 1: "ALS-linked loss of Cyclin F function affects regulatory HSP90 ubiquitination") and the abstract (line 41: "... Together, our results point to a possible Cyclin-F loss-of-function-mediated chaperone dysregulation that might be relevant for ALS..") as suggested.

3- The major strength of this manuscript lies in its comprehensive, detailed and quantitative MS analysis. I therefore am supportive of publication assuming these datasets are made available, which was not clear from the current submission.

All proteomics data has been added as supplementary tables (Table 1-4). In addition, the raw MS data was also uploaded to the PRIDE depository server. Login details are provided.

4- The ubiquitination difference for HSP90AB1 in 4B is not that convincing. The blot is "smudgy" and the difference appears minimal. This should be addressed with a better blot and would be supported by cyclin F overexpression. Along those same lines, the K69 ubiquitin antibody is not validated and is therefore not convincing. Showing that the antibody does not detect a band in cell expressing a K69A mutation, would strengthen this reagents confidence. It is unclear why this antibody would detect a single band and not a smear of polyubiquitination, as might be suggested by 4B.

We fully agree and exchanged the immunoblot in the old Figure 4B with a shorter exposure. In addition, we quantified three biological replicates of TUBE experiment and now provide the statistical analysis next to the pulldown (revised Figure 4C). This updated analysis supports our initial conclusion that HSP90AB1 ubiquitination is at least partially dependent on Cyclin-F. As suggested, we also performed an overexpression experiment and monitored HSP90AB1 ubiquitination using denaturing IPs. While HSP90AB1 protein levels were not affected by proteasomal inhibition through Bortezomib (Btz), this treatment resulted in a robust ubiquitination of HSP90AB1. Consistent with the role of a cullin RING E3 ligase complex (such as SCF^{Cyclin F}) in mediating this ubiquitination, we observed a strong reduction of the ubiquitin signal on HSP90AB1 following MLN4924 treatment. This finding is added as new Figure 4B. The experiment with the HSP90AB1 K69 antibody was performed with the supernatant of a newly established antibody-producing monoclonal hybridoma cell line. Following re-cloning and expansion of that cell line, neither the new supernatant nor the purified antibody detected K69 ubiquitinated HSP90AB1 anymore. Since we also did not have any leftovers from the initial supernatant, we removed the old Figure 4C and 4D which were both related to the HSP90AB1 K69 ubiquitin experiment.

5- Even on long exposure, there appears to be more HSP90AB1 in the input and pulldown of CCNF KO cells in 4F. That is a very interesting finding, but interpretation is complicated by this discrepancy, since increased binding to other factors could be due to this difference. Can this be repeated in conditions where those differences are normalized?

We thank the reviewer for raising this important issue. We quantified the biological triplicates that we had performed and now display the statistical analysis as bar graph next to the immunoblot panel (revised Figure 4E). Importantly, upon normalization to

immunoprecipitated HSP90AB1 (bait), we observed significantly increased binding of the client HSF1 and the co-factor CHIP in the absence of Cyclin-F.

6- How many times were individual experiments in figures 3 and 4 performed?

Binding of wild-type Cyclin-F to HSP90AB1, STIP1 and DNAJC7 was observed in 6, 3 and 2 individual experiments, respectively (Figure 3B and 3C). Binding of HSP90AB1 to different Cyclin-F variants was observed in 3 individual experiments (Figure 3D). The denaturing IPs of HSP90AB1 in Figure 4A and 4B were performed 4 respective 2 times. The TUBE pulldown was repeated in 3 biological replicates and has now been quantified (revised Figure 4C). The HSP90AB1 abundance WB (Figure 4D) was performed 2 times and the HSP90AB IP was repeated 3 times (for ULK1, HSF1, TBK1 and CHIP) and has now been quantified (revised Figure 4E and S4B). For DNAJC7 this experiment has been done twice.

Reviewer #2 (Comments to the Authors (Required)): In the manuscript entitled "ALS-relevant loss of Cyclin F function affects regulatory HSP90 ubiquitination," Siebert and colleagues take a wide systems-level approach to determine possible SCF-cyclin F substrates and interacting partners. They perform most of their experiments as a compare and contrast with wild type cyclin F and mutants of cyclin F that have been linked, genetically, to ALS. They perform a global analysis of ubiquitination site changes in neural-like cell models lacking cyclin F and ALS patient-derived lymphoblastoid cell lines with two separate cyclin F mutations. The authors then go on to perform both traditional affinity based and APEX-based proximity interaction proteomics analysis of wild type cyclin F. From these global analyses, the authors focus on the interaction with cyclin F and HSP90AB1. The authors attempt to demonstrate a specific interaction with cyclin F and HSP90AB1 as well as document that ubiquitination of HSP90AB1 is altered in cells lacking cyclin F. Lastly, they document a modest change in HSP90AB1 association with HSF1 and CHIP/STUB1 in cyclin F knockout cells.

Combined, the studies are interesting, and the larger datasets will be useful additions to researchers interested in cyclin F regulation. However, some critical controls are missing and the authors, in the end, identify two interactions with highly abundant proteins (HSP90 and tubulin) as being specific and possibly informing on cyclin F function with regards to contributions to ALS-phenotypes. However, additional data supporting these interactions as being specific or regulatory (as the title suggests) is needed. Further, additional data is needed to state that cyclin F-mediated ubiquitination of HSP90 regulates HSP90 function. As currently constructed, the data in the manuscript does not support the stated results, nor do they support the conclusion described in the title of the manuscript. I would recommend adding additional data addressing the concerns detailed below or substantially altering the stated conclusions.

Major concerns:

1) The authors need to provide supplemental tables documenting all proteomics data (maxquant outputs) for the ubiquitinome and interaction analyses. These tables should list all identified proteins/sites for all replicates and secondary tables listing the proteins that were selected as specific and of interest based on their criteria. These tables are crucial for others to evaluate the primary data and to supplement their own studies without the need to replicate the experiments.

All proteomics data has been added as supplementary tables (Table 1-4). In addition, the raw MS data was also uploaded to the PRIDE depository server. Login details are provided.

2) Did the authors perform a background proteomics analysis of total proteome changes in their four cyclin F knockout or mutant cell lines? These data are necessary to understand if changes in ubiquitination of specific proteins (up or down) correlate with changes to total protein abundance. It is understood that some of the proteins sampled by the ubiquitin-site enrichment approach may not be observed in a top-level analysis (4000 proteins) of proteome changes. However, for highly abundant proteins like Map1B, ENO1, and ribosomal proteins, their total levels could be easily assessed using already generated samples for ubiquitinome analysis.

We are very grateful for this suggestion. We proteomically analyzed the input samples of the four different diGly IP experiments and provide this data in the new Figure S2B, S2C and in Table2. In addition, this data set was likewise upload to the PRIDE MS database. While Figure S2B gives an overview of the scale of total protein level changes upon Cyclin F knockout or mutation (between 60-70% of the quantified proteins remained unchanged), Figure S2C focuses on proteins with commonly decreased diGly sites in Cyclin F knockout and mutant cells (from Figure 1G). Importantly, the vast majority of potential Cyclin F ubiquitination targets including HSP90AB1 remained unchanged at the protein level upon Cyclin F deficiency.

3) In the end the authors identify 30 overlapping sites comparing their cyclin F KO to mutant lines. Is this overlap significant? What is expected by chance given the number of sites that are observed in all datasets? A similar analysis is needed for the overlapping interacting proteins depicted in 2E.

While we have not done the statistics on this overlap, from a recent parallel study on ALS-linked mutant UBQLN2 (Strohm et al., 2022; PMID: 35777956) we know that the intersection of different patient-derived cells and engineered cell lines can be extremely small (in the case of UBQLN2 this was only one protein, MAP1B). Notably, the number of proteins detected in this study was considerably larger. Along these lines, 30 sites in 22 non-redundant proteins (from Figure 1G) represents a substantial overlap. The same is true for the overlap of interaction and proximity proteomics which contains 103 proteins (Figure 2F). The fact that we could validate one of the hits from these two tool pools (diGly and association) proves that our approach is validate implicating that this data set might contain a number of additional Cyclin-F targets.

4) I appreciate that the authors utilized a control protein (TBC1D7) for their affinity-interaction proteomics to compare against cyclin F. However, the blots that show expression levels of the two proteins (figure 2A) are not done together so it is impossible to judge the relative expression levels of the two proteins (is cyclin F expression 20x as much as TBC1D7?). An HA immunoblot with both cell lines on the same blot is required to make this comparison. Differences in expression levels can have dramatic effects on identified interacting proteins (like chaperones). Indeed, there appears to be a large skew towards cyclin F interacting proteins in figure 2D (SH-SY5Y).

We agree that this is an important point. We revised Figure 2A which now shows Cyclin F and TBC1D7 on the same immunoblot for N2a and SH-SYS5 cells (revised Figure 2A). This

side-by-side comparison shows that Cyclin F is actually expressed to lower amounts than TBC1D7.

5) Despite having a control protein for the affinity interaction studies, the authors lack appropriate controls for their APEX-based proximity studies. They simply compare with and without biotin phenol, which is a fine negative background control, but a positive specificity control is needed (like APEX-TBC1D7). Again, the authors identify many putative interactors in their APEX study (impossible to judge exactly how many as the datasets are not provided) many of which (like UBAP2L) are likely non-specific and would be identified across control samples with APEX-GFP, or other appropriate controls.

As suggested, we performed an additional proximity proteomics experiment comparing APEX2-CCNF and APEX2-TBC1D7 (as negative control) in the presence of biotin-phenol (BP) and H₂O₂. Upon statistical analysis (Student's t-test), we identified 504 proteins that were significantly enriched in the proximity of CCNF (t-test difference > 0.75, FDR corrected, q-value < 0.05) compared to 538 proteins that were detected in the presence of BP in the APEX2 -/+BP experiment. This data is now included as new Figure S2B and Table 4. Looking at known Cyclin F interactors, the new approach did not obviously outperform the APEX2-CCNF -/+ BP analysis as shown in the new Figure S2C. In fact, some of the Cyclin F candidate interacting proteins validated biochemically in our work such as HSP90AB1, STIP1 and TRIM28 were not detected in this additional proximity proteomics experiment. Nevertheless, from the 103 proteins that were commonly detected by proximity (-/+ BP) and interaction proteomics, the comparison of APEX2-CCNF vs APEX2-TBC1D7 confirmed 54 % of these candidates. Hence, this additional data set certainly helped to increase the robustness of our proteomics analysis.

6) The authors focus on the interaction with cyclin F and HSP90AB1 and nicely show specific binding (compared to FBXO28). This data is as consistent with cyclin F being a folding client of HSP90AB1 as opposed to the authors suggestion that cyclin F is regulating HSP90 function. The authors need to state this as a possible likely scenario (given cyclin F overexpression).

This is a valid point. We added a sentence pointing to these two possible scenarios (line 229: "Notably, the HSP90AB1-Cyclin F interaction could represent a ligase-substrate or chaperone-client relationship.").

7) The data describing changes in HSP90 ubiquitination in response to changes in cyclin F is not convincing and more data would have to be collected/shown to make any conclusive statements regarding SCF-cyclin F dependent ubiquitination of HSP90. The data shown in figure 4A could be due to any ligase ubiquitinating overexpressed HSP90AB1. Do endogenous IPs of HSP90 show similar levels of higher-order ubiquitination? Further the TUBE enrichments in cyclin F KO and controls do not reveal convincing changes in apparent HSP90AB1 ubiquitination. These experiments would need several replicates to make any argument that the subtle changes observed are meaningful and reproducible. Do they observe the same changes in cyclin F N2A KO cells or the LCL cyclin mutant cells? The authors highlight many ubiquitination sites on HSP90AB1 in figure 4C. How many of these sites have been observed in other diGLY-based ubiquitinome studies (of which there are many now)? Do these sites increase or decrease in the presence of proteasome inhibitors? Do they increase or decrease in response to Nedd8 E1 inhibition (very relevant for a possible cyclin-F target)? The authors could easily

mine these datasets to strength their argument that these HSP90 ubiquitination events may be regulatory in nature. Amazingly, the authors suggest that they have a HSP90_K69 ubiquitin site specific antibody and show that this immunoreactivity is reduced in cyclin F knock out cells. However, no validation data showing this antibody is indeed specific to K69 ubiquitinated HSP90 is provided (maybe this was done in another study that could be referenced here). Without this validation data, how is the reader to know that this antibody is specific. This is particularly important because there is a paucity of ub-site specific antibodies, and this would be a noteworthy finding.

We are thankful for this constructive criticism. To substantiate our findings, we performed the following additional experiments: Firstly, we exchanged the immunoblot in the old Figure 4B with a shorter exposure. In addition, we quantified three biological replicates of this experiment and now provide the statistical analysis next to the TUBE pulldown which shows a significant reduction of ubiquitinated HSP90AB1 upon Cyclin F knockout (revised Figure 4C). Due to the unavailability and delivery delay of commercial TUBE reagents we prioritized this experiment in SH-SY5Y cells over similar ones in N2a and LCLs. Secondly, we examined the levels of ubiquitinated HSP90AB1 in cells grown in the absence and presence of Bortezomib or/and MLN4924. While HSP90AB1 protein levels were not affected by proteasomal inhibition, this treatment resulted in a robust ubiquitination of HSP90AB1. Consistent with the role of a cullin RING E3 ligase complex (such as SCF^{Cyclin F}) in mediating this ubiquitination, we observed a strong reduction of the ubiquitin signal on HSP90AB1 following MLN4924 treatment. This new finding is added as new Figure 4B. The experiment with the HSP90AB1 K69 antibody was performed with the supernatant of a newly established antibody-producing monoclonal hybridoma cell line. Following re-cloning and expansion of that cell line, neither the new supernatant nor the purified antibody detected K69 ubiquitinated HSP90AB1 anymore. Since we also did not have any leftovers from the initial supernatant, we removed the old Figure 4C and 4D related to the HSP90AB1 K69 ubiquitin experiment. As suggested, we performed endogenous IPs of HSP90AB1, however, under denaturing conditions the antibody was not able to efficiently precipitate HSP90AB1. Mining public PTM databases (e.g., PhosphoSitePlus) revealed that all (12) diGly sites that we detected on HSP90AB1 have been previously found in other cell lines (e.g., Hep2 and Jurkat) by other groups (e.g., Akimov V, et al. (2018) Nat Struct Mol Biol 25, 631-640). Intriguingly, the vast majority of these sites were not sensitive to Btz treatment with the exception of HSP90AB1 K69 which increased upon proteasomal inhibition. This finding is consistent with our observation that HSP90AB1 is heavily ubiquitinated upon Btz treatment as shown in the new Figure 4B. Importantly, this ubiquitination event does not seem to be a signal for proteasomal degradation since we did not observe any overt changes in protein abundance of HSP90AB1 in parental and CCNF KO cells grown in the absence or presence of Btz (Figure 4D). We corroborated this finding by cycloheximide chase assays monitoring endogenous or overexpressed HSP90AB1. This additional data is now provided as new Figure S4A. Mining of a MLN4924 diGly dataset (e.g., Kim W, et al. (2011) Mol Cell 44, 325-40) did only reveal one of the sites that we detected (K435). While Kim et al. did not find evidence for a cullin RING ligase (CRL) dependence for this particular site in their large-scale approach, we clearly detect such a dependence when we perform denaturing IPs from cells grown in the absence and presence of Btz and/or MLN4924 as shown in the new Figure 4B.

Minor concerns:

1) In the figure legend for figure 1 and in the text describing the results the author say, "Using statistical analysis (one-sample t-test with a p-value of < 0.05), we identified between 38 and 100 diGly sites (27-

83 proteins) with log₂ (SILAC) ratios < -0.5 in SH-SY5Y and LCLs as well as log₂ (SILAC) ratios < -0.75 in N2a cells." It is clear that a statistical analysis was done to determine possible differentially regulated sites but those cutoffs aren't discussed in the legend as only log₂ fold change cutoffs are described. Are the sites depicted in 1G only based on Log₂ fold change cutoffs or are these also below a p-value cutoff. Further, it does not appear that the authors have utilized any adjustment for p-values based on multiple hypothesis testing which is critical to do for these types of datasets. I apologize if this was done and I missed it in the methods section.

We apologize that we did not provide this information in the figure legend. In fact, we repeated the statistical analysis and performed a more stringent filtering of diGly sites with log₂ fold change < -0.5 or > 0.5 and a q-value < 0.05 for all cells. This is now clearly indicated in the legend of Figure 1D and 1E, on the revised Figure 1D and 1E as well as in the main text. For Figure 1G, we aimed for identifying robust diGly sites which similarly decreased across the different cell types (N2A, SH-SY5Y and LCLs from two different ALS patients) and CCNF deficiencies (KO vs. mutation). To account for this heterogeneity, we lowered the threshold and only filtered diGly sites based on log₂ fold change < -0.5. Hence, the stringency of this selection does not stem from the strongest decrease within a particular dataset but rather from the fact that the decrease is observed in very different conditions. For the interaction and proximity proteomics, we also updated the analysis of HA-IP and APEX 2 in N2a and SH-SY5Y and now present FDR corrected q-values < 0.05 (revised Figure 2E and 2F).

2) The authors describe some characterization studies on the S621G cyclin F mutant but then never use that same mutant in their studies. Have similar studies been done on the D628V or V335M mutants? Because the authors never use the S621G mutant, why spend so much time describing previous experiments with that mutant?

At the time of submission, detailed studies with ALS-linked mutant Cyclin F were focused on this particular mutant. To live up to this reference, we thought to introduce the literature for S621G cyclin-F properly.

Reviewer #3 (Comments to the Authors (Required)):

Siebert et al. aim to elucidate the role of CCNF mutations in amyotrophic lateral sclerosis 1) in the substrate adaptor function of cyclin F and 2) if alterations in SCFcyclin F mediated ubiquitination are of pathophysiological relevance in patients with CCNF mutations. To date, several reports characterizing the consequences of the S621G CCNF mutation in ALS have been published, however ubiquitination targets and pathophysiological effects of other CCNF mutations in ALS still remain unclear.

Main points:

The authors focus on uncovering ubiquitination targets of cyclin F in two CCNF knockout cells as well as patient-derived lymphoblastoid cell lines. In Fig1A CCNF blots consistently show double bands. However, in SH-SY5Y and the LCL cells the upper band is marked as CCNF, whereas the N2a WT cells seem to only show the lower <100kDa band, which is lost upon KO of CCNF, whereas the upper band does not appear in neither WT nor KO cells. Are there different isoforms of CCNF in respective cell lines? How do the authors explain this difference?

The antibodies that are available for Cyclin F only work very poorly on mouse cell lines and detect different unspecific background bands in N2a and SH-SY5Y cells. Human and murine Cyclin F differ in size only by ~10 amino acids (based on Uniport entries) but certainly they do migrate at slightly different heights in SDS-PAGE (as show in Figure 1A and 1B), possibly reflecting distinct modifications or alternative splice variants. This would be an interesting aspect to explore in future studies.

What was the reasoning behind going forward with the SH-SY5Y cell line regarding HSP90AB1 binding and ubiquitination and not N2a cells and especially mutated LCLs vs. controls?

For the follow-up experiments, we selected the human neuron-like SH-SY5Y as we felt these cells are most relevant for ALS. Additionally, the patient LCLs grew in suspension and extremely slowly, making them harder and more time consuming to handle. Certainly, experiments in primary neurons or iPSCs would be very insightful but in the respect of time we will tackle these in the next project.

And similarly, did the authors proceed to perform ubiquitination analyses of HSP90AB1 only with SH-SY5Y CCNF WT and KO cells or also V335M and D628V mutations, if those mutations were of special interest in this particular study?

For addressing the ubiquitination status of HSP90AB1, we indeed only performed experiments in SH-SY5Y cells. Time constraints and availability of reagents due to Corona were the biggest factors which prevented us from expanding our analysis in Cyclin F mutant cells.

Minor points:

There is no mention of Fig S1 in the text.

We apologize for that and made sure that all main and supplementary figures are mention in the text.

Please provide information regarding ethics approval for patient-derived materials to the methods section.

This information has now been provided in the Material & Method section ("For the collection and use of blood cells from ALS patients as well as for whole exome sequencing of blood DNA, written informed consent was obtained from all individuals. The experiments have been approved by the local ethical committees of the Medical Faculties Ulm (Ulm University) and Mannheim (ethical committee II of the University of Heidelberg). Approval numbers are Nr. 19/12 and 2020-678N, respectively."). In addition, we added the following paragraph to our Material & Method section: "The LCL line with p.D628V mutation was derived from a male patient with spinal onset of familial ALS at the age of 47 years. Both his father and paternal grandfather were affected by the disease, in agreement with an autosomal-dominant mode of inheritance. The patient did not suffer from FTD comorbidity. Due to loss of follow-up, the survival status of the patient is unknown. The LCL line with the p.V335M mutation was derived from a female ALS patient without a family history for the disease. She also had a spinal onset of disease at the age of 62 years with distal extensor

weakness in the lower extremities, followed by paresis in the upper extremities and subsequently bulbar symptoms. She had clinical signs of both upper and lower motor neuron degeneration. Sensory function and coordination were unremarkable. Both patients were subject to whole exome sequencing, and genetic variants in other known ALS disease genes were excluded."

August 9, 2022

RE: Life Science Alliance Manuscript #LSA-2021-01359R

Prof. Christian Behrends
Ludwig-Maximilians-University Munich
Munich Cluster for Systems Neurology
Feodor-Lynen Strasse 17
Munich, Bayern 81377
Germany

Dear Dr. Behrends,

Thank you for submitting your revised manuscript entitled "ALS-linked loss of Cyclin-F function affects regulatory HSP90 ubiquitination". We would be happy to publish your paper in Life Science Alliance pending final revisions necessary to meet our formatting guidelines.

- please tone down title and abstract according to Reviewer 1' remaining comments
- please provide your manuscript text in editable doc file format
- please add a summary blurb/ alternate abstract in our system
- please add the Twitter handle of your host institute/organization as well as your own or/and one of the authors in our system
- please use the [10 author names, et al.] format in your references (i.e. limit the author names to the first 10)
- please add your table legends to the main manuscript text

A. FINAL FILES:

B. MANUSCRIPT ORGANIZATION AND FORMATTING:

Sincerely,

Reviewer #1 (Comments to the Authors (Required)):

Summary: The SCF family of cullin ring ligases play critical roles in normal cell physiology and many are implicated in disease progression. SCF ligases use interchangeable substrate receptor F-box proteins to designate substrates for ubiquitination and degradation. Cyclin F is the founding member of the F-box family and plays important roles in cell cycle. Recently, mutations in cyclin F were identified in patients with ALS, although it remains unclear how these mutations contribute to disease. Determining these mechanisms will shed light on disease pathogenesis. The authors here undertake a comprehensive analysis of proteome wide changes, using both cyclin F KO cell lines and those reconstituted with mutant proteins based on those observed in ALS patients. They combine this with traditional APMS, proximity labeling MS and ubiquitin proteomics. Together, these data provide a comprehensive snapshot of cyclin f regulated proteomes.

The authors took steps to address several of my concerns. However, I still have reservations regarding the ubiquitination of HSP90AB1. It was detected as being different in the MS experiments that were performed, but the differences show by blot is difficult to see. Moreover, the differences in binding depicted in 4E are still confounded by differences in expression of the protein being precipitated. Because of this, I think the title of the paper overstates the findings in the manuscript current form. The significance of this ubiquitination is also overstated in the abstract. If those were to be modified I would be supportive of its publication without further experiments or review.

Reviewer #2 (Comments to the Authors (Required)):

I thank the authors for their thoughtful consideration of reviewers previous concerns. They have successfully addressed all of my previous concerns and I support publication going forward.

Reviewer #3 (Comments to the Authors (Required)):

The revised manuscript presented by Siebert and Weishaupt addressed all points raised by this reviewer. No further comments.

September 5, 2022

RE: Life Science Alliance Manuscript #LSA-2021-01359RR

Prof. Christian Behrends
Ludwig-Maximilians-University Munich
Munich Cluster for Systems Neurology
Feodor-Lynen Strasse 17
Munich, Bayern 81377
Germany

Dear Dr. Behrends,

Thank you for submitting your Research Article entitled "ALS-linked loss of Cyclin-F function affects HSP90". It is a pleasure to let you know that your manuscript is now accepted for publication in Life Science Alliance. Congratulations on this interesting work.

DISTRIBUTION OF MATERIALS:

Again, congratulations on a very nice paper. I hope you found the review process to be constructive and are pleased with how the manuscript was handled editorially. We look forward to future exciting submissions from your lab.

Sincerely,
